



# An Estimate of Global, Regional and Seasonal Cirrus Cloud Radiative Effects Contributed by Homogeneous Ice Nucleation

David L. Mitchell[1], John Mejia[1], Anne Garnier[2], Yuta Tomii[1], Martina Krämer[3,4] and Farnaz Hosseinpour[1]

[1]Desert Research Institute, Reno, 89512-1095, USA

[2]Science Systems and Applications, Inc., Hampton, Virginia, USA

[3]Institute for Energy and Climate Research (IEK-7), Research Center Jülich, Jülich, Germany

[4]Institute for Atmospheric Physics (IPA), Johannes Gutenberg University, Mainz, Germany

*Correspondence to*: David Mitchell (david.mitchell@dri.edu)

**Abstract.** There are two fundamental mechanisms through which cirrus clouds form; homo- and heterogeneous ice nucleation (henceforth hom and het). The relative contribution of each mechanism to ice crystal production often determines the microphysical and radiative properties of a cirrus cloud. This study attempts to estimate the radiative contribution of hom relative to het by constraining the cloud microphysics in a climate model to conform with satellite retrievals of cirrus cloud effective diameter $D_e$, where the sampled cirrus cloud base had a temperature T < 235 K (-38 °C). The CALIPSO (*Cloud-Aerosol Lidar and Infrared Pathfinder Satellite Observation)* satellite retrievals for cirrus clouds are compared against an updated in situ cirrus cloud property climatology to evaluate similarities and differences. In this climate modeling study, we ask how the cloud radiative effect (CRE) based on retrieved cirrus cloud properties compares with the CRE predicted by a model configuration representing cirrus clouds formed only through het and also with the CRE predicted by the standard configuration of the model. To answer this question, we constrained version 2 of the Morrison-Gettelman cloud microphysics scheme (MG2), which is used in several climate models, using effective diameter ($D_e$) retrievals from the CALIPSO satellite. A new subroutine within the MG2 scheme provides retrieved $D_e$ as a function of temperature (T), latitude, season and land fraction, while ice particle mass and area relationships are used to relate $D_e$ to the ice particle size distribution (PSD) slope and to produce new relationships for the number- and mass-weighted ice fall speeds. These and other modifications rendered the MG2 microphysics consistent with the $D_e$ constraint. Using 40-year simulations of the Whole Atmosphere Community Climate Model version 6 (WACCM6), the CRE outside the tropics from the $D_e$-constrained WACCM6 was greater than standard WACCM6 by 1.63 W m$^{-2}$ in the Northern Hemisphere (NH) and 2.59 W m$^{-2}$ in the Southern Hemisphere (SH). Using the version of WACCM6 designed to represent cirrus clouds formed only by het (instead of using standard WACCM6), this difference was 2.37 W m$^{-2}$ in the NH and 2.55 W m$^{-2}$ in the SH. These differences are larger when only non-summer months are considered.





## 1  Introduction

Cirrus clouds contain only ice particles (i.e. no liquid cloud droplets), a condition guaranteed when cloud temperatures (T) are less than ~ -38° C.  Their ice crystals can form by either of two mechanisms; homogeneous or heterogeneous ice nucleation (henceforth hom and het).  The former requires no ice nucleating particles (INP) and

can proceed through the freezing of haze and cloud solution droplets when $T \leq 235$ K (-38° C) and the relative humidity with respect to ice (RHi) exceeds some threshold where RHi > ~ 145% (Koop, 2000).  This results in relatively high concentrations of ice particles (N), where in situ measurements of N are typically > ~ 200 $L^{-1}$ (liter$^{-1}$) whereas het generally produces N < ~ 200 $L^{-1}$ (Barahona and Nenes, 2009; Jensen et al., 2012a, b; Cziczo et al., 2013).  Under very cold and unique conditions of weak and relatively short-lived updrafts (e.g. low-amplitude

gravity waves), N resulting from hom may be < 100 $L^{-1}$ (Sprichtinger and Krämer, 2013; Krämer et al., 2016), and under atypical conditions (such as high concentrations of mineral dust), N resulting from het can exceed 200 $L^{-1}$.  In cirrus clouds, het may occur at any RHi > 100%, and in the context of a cloud parcel moving in an updraft, ice is first produced through het, and subsequently through hom if the het-produced ice crystals do not prevent the RHi from reaching the threshold RHi needed for hom to occur (e.g. Haag et al., 2003).  Overall, cirrus clouds formed

primarily through hom will likely have substantially higher N and smaller effective diameters ($D_e$) relative to cirrus formed primarily through het.  For a given ice water content (IWC) or ice mass mixing ratio ($q_i$) and temperature range, these two types of cirrus clouds will therefore display considerably different radiative properties, making it critical to properly attribute hom and het in climate modeling.

Current climate models capable of predicting hom and het in cirrus clouds assume that ice nucleation occurs in the

presence of pre-existing ice crystals (i.e. the pre-existing ice assumption; Shi et al., 2015). This strongly favors het over hom since the pre-existing crystals may exhibit considerable ice surface area (relative to new ice crystals) and prevail in the competition for water vapor, preventing the RHi from reaching the RHi threshold for hom.  This assumption is universally applied under all conditions.

However, it is possible that hom plays a larger role in cirrus clouds than currently predicted in climate models.  For

example, the satellite remote sensing study by Zhao et al. (2018) demonstrated hom had a strong impact on cirrus cloud microphysics under relatively clean (relatively low aerosol optical depth) conditions.  Another satellite remote sensing study by Sourdeval et al. (2018) shows that N in cirrus clouds (T < 235 K) outside the tropics is relatively high during the winter season, with relatively low N for T > 235 K.  This same result (highest N during winter) was observed in the cirrus cloud remote sensing studies of Mitchell et al. (2016; 2018), who suggested that hom was

more active during winter due to a reduction of convective mixing of ice nucleating particles (INP) from the surface to cirrus cloud levels.  Anvil cirrus, most common during summer, contain advected pre-existing ice that may also enhance het.  These studies also found N outside the tropics (± 30° latitude) was higher over mountainous terrain,



attributing this to mountain-induced wave clouds having relatively strong and sustained updrafts (and thus greater cooling rates producing high supersaturations) conducive for hom. This was also observed in the satellite remote sensing study of Gryspeerdt et al. (2018) that explained the higher N over mountainous terrain in a similar way. In both studies N was highest near cloud top, which appeared to directly affect N at lower levels.

In addition, Mitchell et al. (2016; 2018) found that N was relatively high at high latitudes. This was explained by the relatively pristine conditions associated with relatively low concentrations of INPs, especially over the Southern Ocean (Vergara-Temprado et al., 2018; McCluskey, 2018). That is, when N produced by het is relatively low, the ice surface area produced is often inadequate to prevent the RHi from climbing and reaching the hom threshold in a cirrus cloud updraft, resulting in higher N produced by hom. Similar results over the Southern Ocean are shown in

Figure 1 of Gryspeerdt et al. (2018).

Motivated by these findings, an experiment was designed to investigate the radiative impact of hom (relative to het) on spatial and temporal (i.e. seasonal) scales. The experiment combined CALIPSO (*Cloud-Aerosol Lidar and Infrared Pathfinder Satellite Observation*) satellite retrievals of effective diameter ($D_e$) with climate modeling, where $D_e$ was used to constrain the cloud microphysics treatment in the climate model. The climate model was

constrained in two ways; one based on global $D_e$ retrievals while the other was based on $D_e$ characteristic of het conditions. Differences between these two constrained versions are expected to reveal the impact of hom.

Specifically, $D_e$ is used to constrain version 2 of the Morrison-Gettelman cloud microphysics scheme (MG2; Gettelman and Morrison, 2015; Gettelman et al., 2015), used in the Community Atmosphere Model version 6 (CAM6), the Whole Atmosphere Community Climate Model version 6 (WACCM6), and version 2 of the

Community Earth System Model (CESM2). For this study, WACCM6 is used to capture potential changes to the stratosphere. The methodology in Section 2 describes how $D_e$ is used to determine the ice particle size distribution (PSD) slope and the number- and mass-weighted ice fall speeds, as well as other aspects. The experimental design is described in Section 3, followed by a comparison in Section 4 between the $D_e$ retrievals and a recently developed climatology of cirrus cloud ice particle properties. Section 5 contains the results and discussions thereof, and

conclusions are given in Section 6.

## 2   Methodology

A CALIPSO satellite remote sensing method designed for cirrus clouds uses co-located measurements from the Infrared Imaging Radiometer (IIR) and the CALIOP (Cloud and Aerosol Lidar with Orthogonal Polarization) lidar aboard CALIPSO to retrieve N by exploiting the sensitivity of the effective absorption optical depth ratio, $\beta_{eff}$, to

small ice crystals, where this ratio is based on the 10.6 and 12.05 µm IIR channels. Also retrieved are $D_e$, ice water content (IWC), ice water path (IWP) and cloud visible optical depth (OD) for single-layer cirrus clouds having 0.3 ≤




OD ≤ 3.0, where cloud base having a temperature ≤ 235 K is always detected. This retrieval is described in Mitchell et al. (2016; 2018) where two years are evaluated to analyze the dependence of cirrus cloud N and $D_e$ on altitude, temperature, latitude, season and surface type (i.e. ocean vs. land, or land-fraction). This two-year data set is used in this study to constrain $D_e$ in the WACCM6 that uses the MG2 scheme. That is, six 30° latitude zones, four seasons

and ocean vs. land designations were used to produce 48 $D_e$-T relationships defined in look-up tables (LUTs). These $D_e$ LUTs are contained within a subroutine that is called by the MG2 scheme to produce a cirrus cloud $D_e$ for a given T, latitude, season and land-fraction. From this $D_e$ are calculated the cirrus PSD slope $\lambda$ and corresponding number- and mass-weighted ice fall speeds $V_N$ and $V_m$. The CALIPSO retrieval's sensitivity to N renders median $D_e$ as small as 11 μm, resulting in very narrow PSD under some conditions where $V_N$ and $V_m$ are not predicted well

by the standard MG2 scheme. For this reason and others we developed new expressions for $V_N$ and $V_m$.

### 2.1 Calculating the cirrus PSD slope parameter from $D_e$

To constrain the MG2 scheme with $D_e$ retrievals requires modifying the MG2 physics to ensure that the physics is compatible and consistent with this constraint. This means that $\lambda$ must be derived from the retrieved $D_e$, beginning with the PSD growth stage at ice nucleation and sustaining this $\lambda$ thereafter. This was done by calculating $\lambda$ from $D_e$

by inverting equation (14) in Erfani and Mitchell (2016; henceforth EM2016):

$$\lambda = \left[ \frac{2\, \rho_i\, \gamma\, \Gamma(\delta+\nu+1)\, D_e}{3\, \alpha\, \Gamma(\beta+\nu+1)} \right]^{[1/(\delta-\beta)]} \tag{1}$$

where $\Gamma$ denotes the gamma function, $\rho_i$ = bulk density of ice (0.917 g cm-3), $\nu$ = PSD dispersion parameter = 0 in the MG2 scheme, and $\alpha$, $\beta$, $\gamma$ and $\delta$ are the ice particle mass (m) and projected area (A) coefficients in their respective power laws:

$$m = \alpha\, D^\beta , \tag{2}$$

$$A = \gamma\, D^\delta , \tag{3}$$

where D is the ice particle maximum dimension. Unfortunately, these coefficients, $\alpha$, $\beta$, $\gamma$ and $\delta$, are weakly dependent on D, and Eq. (1) is only accurate over the D-range associated with $\lambda$ and $D_e$, especially at relatively small D. To account for this non-linearity, m and A are calculated as described in EM2016:

$$\ln m = a_0 + a_1 \ln D + a_2 (\ln D)^2 , \tag{4}$$

$$\ln A = b_0 + b_1 \ln D + b_2 (\ln D)^2 , \tag{5}$$



where $a_0$, $a_1$, $b_0$ and $b_1$ are constants. Note that ln denotes natural log and m, A and D are in g, cm$^2$ and cm. See Tables 1 and 2 in EM2016 for the coefficients to apply, using the temperature interval for -40 to -55 °C and synoptic cirrus clouds. Note that using curves for other temperature intervals and/or anvil cirrus makes little difference in these relationships since they are similar. Note that the PSD mean ice particle size, $D_{mean}$, is related to $\lambda$ as $D_{mean} =$

$(\nu + 1)/\lambda$ for an analytical gamma size distribution given as:

$$N(D) = N_0\, D^\nu \exp(-\lambda D), \tag{6}$$

where $N_0$, $\nu$ and $\lambda$ are constants. As described in EM2016, an iterative solution employing Eqns. (4), (5) and (1) yields an accurate relationship between $D_{mean}$ and $D_e$, and thus $\lambda$ and $D_e$. This relationship is shown by the orange circles in Fig. 1, and a curve-fit to this solution is given by this equation (black curve in Fig. 1):

$$\ln(D_{mean}) = -8.727\,(-0.1962\,\ln(D_e) + 1.0)^{1/2} + 7.680, \tag{7}$$

where $D_{mean}$ and $D_e$ are in micron units. Although the EM2016 ice particle mass and projected area relationships (i.e. Eqns. (4) and (5)) are technically valid down to D = 20 μm, these relationships were extrapolated down to $D_{mean}$ = 1 μm to produce Eq. (7). In doing so, they were constrained to not exceed the value of an ice sphere ($\rho_{ice}$ = 0.917 g cm$^{-3}$) having the same D. This affected $D_{mean}$ < 27 μm for projected area and $D_{mean}$ < 10 μm for mass. These

relationships are based on synoptic cirrus clouds for -55°C < T < -40°C, given in Table 2 of EM2016. Similar m-D and A-D relationships are given in EM2016 for other temperature intervals as well as for anvil cirrus clouds, but there was generally not much variation among these relationships. Therefore, the m-D/A-D relationships used here are considered representative for all cirrus clouds. These relationships were utilized in a manner similar to the way they were used to produce Fig. 13 in EM2016. Specifically,

$$\beta = a_1 + 2\,a_2\,\ln D_m , \tag{8}$$

$$\alpha = \exp[a_0 + a_1 \ln D_m + a_2\,(\ln D_m)^2]/\,D_m^{\,\beta} , \tag{9}$$

where $D_m$ is the median mass dimension, given for a gamma PSD as

$$D_m = (\beta + \nu + 0.67)/\lambda. \tag{10}$$

The equations giving $\delta$ and $\gamma$ are analogous to Eqns. (8) and (9), but use different constants that are given in Table 2

of EM2016 for synoptic cirrus clouds, -55°C < T < -40°C, and are based on $D_A$, the median area dimension, given in EM2016 as

$$D_A = (\delta + \nu + 0.67)/\lambda. \tag{11}$$



To generate the data in Fig. 1, $D_{mean}$ was incremented, and new values for α, β, γ and δ were calculated as above for each $D_{mean}$ for use in Eq. (1) to solve for $D_e$. Moreover, $D_e$ is commonly defined as

$$D_e = (3/2)\, IWC/(\rho_i\, A_{PSD}) , \tag{12}$$

where IWC and $A_{PSD}$ are the PSD ice water content and projected area, respectively, and $\rho_i$ is the bulk density of ice
(Mitchell, 2002). Due to its dependence on IWC, $D_m$ is used to calculate α and β in Eqns. (8) and (9), and due to its dependence on $A_{PSD}$, $D_A$ is used to calculate γ and δ in this iterative solution for $D_e$. This is the most accurate way we know of for calculating the relationship between $D_e$ and $D_{mean}$, especially at small values of $D_e$ that correspond to the smaller retrieved values of $D_e$. If $D_{mean} > 3$ mm (3000 μm), then $D_e = 166$ μm since Eq. (7) asymptotes to this value. Since (7) may not be valid for $D_{mean} > 3000$ μm, for cirrus, $D_{mean} \leq 3000$ μm in this modified MG2 scheme.

## 2.2 Calculation of ice particle fall speeds

Two PSD slope values are used in the MG2 scheme, one for cloud ice ($\lambda_i$) and one for snow ($\lambda_s$). Since we are forcing predicted cirrus clouds in MG2 to conform with CALIPSO retrievals of $D_e$, both $\lambda_i$ and $\lambda_s$ have the same value corresponding to the retrieved $D_e$. The number- and mass-weighted ice fall speeds, $V_N$ and $V_m$, can be expressed as a function of $D_{mean}$ and $D_e$, respectively, over the entire range of observed $D_{mean}$ and $D_e$ in cirrus clouds
as shown in Mishra et al. (2014). Thus, if $V_N$ and $V_m$ are formulated in this way, it is not necessary to have separate expressions for $V_N$ and $V_m$ for cloud ice and snow, which is the current practice in the MG2 scheme. Therefore, in this modified treatment of cirrus clouds, we have a single expression for $V_N$ as a function of $D_{mean}$, and a single expression for $V_m$ as a function of $D_e$.

We use the same definition of $V_N$ and $V_m$ as found in the MG2 scheme:

$$V_N = (\rho_o/\rho)^{0.35}\, a\, \Gamma(b + \nu + 1) / \Gamma(\nu + 1)\, \lambda^b , \tag{13}$$

$$V_m = (\rho_o/\rho)^{0.54}\, a\, \Gamma(b + \beta + \nu + 1)/ \Gamma(\beta + \nu + 1)\, \lambda^b , \tag{14}$$

where $a$ and $b$ refer to the prefactor and exponent of an empirical fall speed power law:

$$V = a\, D^b , \tag{15}$$

$\rho$ is air density and $\rho_o$ is $\rho$ at a reference pressure and temperature of 850 hPa and 273 K. For number weighted
cloud ice, the MG2 scheme uses 0.35 as the power for the ratio $\rho_o/\rho$, but otherwise uses 0.54 based on Heymsfield et al. (2007). Since this fall speed treatment is only for cirrus clouds, an exponent of 0.35 appears most appropriate for $V_N$. The difference with this new approach is that $a$, $b$ and β are constantly updated as a function of ice particle size. This process for β is described above. In Mitchell (1996), $a$ and $b$ are defined theoretically as





$$V = c\,\mu\,\left[\frac{2\,\alpha\,g}{\rho\,\mu^2\,\gamma}\right]^{d}\,D^{\,d(\beta+2-\delta)-1} \tag{16}$$

where $a$ in Eq. (15) is defined by the terms preceding D and $b$ is defined by the power term for D. The terms $c$ and $d$ are constants describing the particle flow regime in Mitchell (1996), but are variables in this treatment that depend on ice particle size and mass as described in Mitchell and Heymsfield (2005). They relate the Best number X with the Reynolds number Re:

$$Re = c\,X^d. \tag{17}$$

When calculating $V_n$, the median number concentration dimension $D_N$ is used to obtain $c$ and $d$ from the Mitchell-Heymsfield (2005) scheme, as well as $\alpha$, $\beta$, $\gamma$ and $\delta$ as described in Sect. 2.1 (analogous to $D_m$ and $D_A$), where $D_N = (\nu +0.67)/\lambda$. When calculating $V_m$, the median mass dimension $D_m$ from Eq. (10) is used to obtain $c$ and $d$ in Eq. (16), as well as for $\alpha$, $\beta$, $\gamma$ and $\delta$ used in Eq. (16). In other words, D in Eq. (16) takes on the value of either $D_N$ or $D_m$. The variables $\mu$ and $\rho$ are kinematic viscosity and air density, and g is the gravitation constant, where $\mu = \eta/\rho$ and $\eta$ is the dynamic viscosity, estimated as:

$$\eta = 2.48 \times 10^{-6}\,(T^{0.754}), \tag{18}$$

with T in degrees Kelvin and $\eta$ is in g cm$^{-1}$ s$^{-1}$. Air density is given as

$$\rho = p / (R_a\,T), \tag{19}$$

where p = pressure in dynes cm$^{-2}$ (equals p in hPa $\times$ 1000), $R_a = 2.867\times10^6$ erg g$^{-1}$ K$^{-1}$ is the gas constant for dry air, T is in K and $\rho$ is in g cm$^{-3}$ (multiply by $10^3$ for kg m$^{-3}$).

While the ice fall speed method of Heymsfield and Westbrook (2010) appears more accurate, the method of Mitchell (1996) yields fall speeds within ~ 10% of those based on Heymsfield and Westbrook (2010) for ice particles found in cirrus clouds (Mitchell et al., 2011). This is because ice particles in cirrus clouds generally have irregular "blocky" shapes with relatively high area ratios (defined as ice particle projected area divided by area of circle having the same maximum dimension), with aspect ratios relatively close to unity. For such ice crystals (T < -38°C), the difference in fall speeds between schemes is minimal (Heymsfield and Westbrook, 2010), ~ 10% or less based on Mitchell et al. (2011). To correct for this overestimation, the fall speeds predicted by (13) and (14) are multiplied by 0.90 to bring them closer to Heymsfield-Westbrook (2010) values.



The result of this methodology for $V_N$ is shown in Fig. 2. The procedure described above generated the orange data points shown, relating $D_{mean}$ to $V_N$. This data was then approximated by a $2^{nd}$ order polynomial curve-fit, shown by the black curve in Fig. 2, having the equation:

$$\ln(V_N) = -6.241 + 2.713 \ln(D_{mean}) - 0.1716 \, [\ln(D_{mean})]^2, \tag{20}$$

where $D_{mean}$ is in microns and $V_N$ is in cm s$^{-1}$. Equation (20) is used in the modified MG2 scheme for both cloud ice and snow since $\lambda_i$ and $\lambda_s$ are identical to force conformity with CALIPSO retrieved $D_e$ (for cirrus clouds only).

The result of this same methodology for $V_m$ is shown in Fig. 3. Again, the data generated from the above methodology are indicated by the orange circles, while the black curve is the curve-fit having the form:

$$\ln(V_m) = -4.484 + 1.809 \ln(D_e), \tag{21}$$

where $D_e$ is in microns and $V_m$ is in cm s$^{-1}$. Both Eqns. (20) and (21) assume reference conditions of p = 850 hPa and T = 273 K, which is also assumed in the MG2 scheme. The linear curve fit in Fig. 3 ignores the first two data points (smallest $D_e$) where $D_e$ is smaller than any of the CALIPSO retrieved $D_e$ values. This was done to reduce computation time while maintaining the same accuracy (since $D_e$ is restricted to the range of retrieved median $D_e$ values). Nonetheless, the following line equation fits all the data points in Fig. 3, with improved accuracy for the
smallest $D_e$:

$$\ln(V_m) = -5.059 + 2.079 \ln(D_e) - 0.03119 \, [\ln(D_e)]^2. \tag{22}$$

Note that Eq. (21) has the simple form

$$V_m = 0.01129 \, D_e^{\ 1.809} . \tag{23}$$

Again, $D_e$ is in microns and $V_m$ is in cm s$^{-1}$, and assumed reference conditions are p = 850 hPa and T = 273 K.

These formulations for $V_N$ and $V_m$ are compared with $V_N$ and $V_m$ determined by other studies in Fig. 4. The ice fall speed schemes compared are from the MG2 scheme, from Mishra et al. (2014) and from Heymsfield and Westbrook (2010). While $V_N$ and $V_m$ are PSD integrated values in the current treatment, in Mishra et al. (2014) and in MG2, the treatment for the Heymsfield-Westbrook scheme is based on $D_{mean}$ and $D_m$, respectively. The Mishra et al. scheme generally agrees within 20% and 30% for $V_N$ and $V_m$, respectively, while the Heymsfield-Westbrook
scheme generally agrees with $V_m$ within 30%.





### 2.3 Calculation of cirrus cloud ice particle number concentration

Using the cloud ice number concentration $N_i$ and $\lambda$ from the last time step, the MG2 microphysics determines the cloud ice water content (IWC), or, in MG2 terms, the mass mixing ratio for cloud ice, and similarly for calculating the mass mixing ratio for snow. In the modified MG2 scheme, $N_i$ and $N_s$ (number concentration for snow) are

determined through the mass balance relationship for the assumed gamma PSD (Mitchell et al., 2006, Eq. 29):

$$N = \frac{IWC \ \lambda^{\beta}}{\alpha \ \Gamma(\beta + 1)} = \frac{\rho_{air} \ q_{ice} \ \lambda^{\beta}}{\alpha \ \Gamma(\beta + 1)} , \tag{24}$$

where $\rho_{air}$ = air density in kg/m$^3$ and $q_{ice}$ is the ice mixing ratio (either for cloud ice or snow) in kg/kg. The MG2 scheme provides an initial estimate of nucleated ice embryos from which an initial $q_{ice}$ is estimated. But thereafter, $\lambda$ is given by Eq. (7) and $N_i$ and $N_s$ are calculated from Eq. (24). In this way, "ice nucleation" is driven by the CALIPSO $D_e$ retrievals via Eq. (24), eliminating most of the uncertainty associated with N ($N_i + N_s$). Ice nucleation

is poorly understood, and this approach is designed to yield a more realistic estimate of N in cirrus clouds. For realistic N values, IWC is not very sensitive to N, with the supersaturation with respect to ice, $S_i$, increasing when N is relatively low and decreasing when N is relatively high. These changes in $S_i$ alter the rate of vapor deposition to ice crystals, reducing the sensitivity of IWC to changes in N.

For estimating $N_i$ and $N_s$ in cirrus clouds, representative values for $\alpha$ and $\beta$ in Eq. (24) are needed regarding the

cloud ice and snow fraction. For the SPARTICUS synoptic cirrus cloud dataset described in Mishra et al. (2014) and Mitchell et al. (2018, henceforth M2018), $D_{mean}$ (in microns) can be very crudely approximated from cloud temperature (°C) using the following linear regression (which works best for T > -42°C):

$$D_{mean} = 5.09 \ T + 324 . \tag{25}$$

Figure 5 shows this SPARTICUS data along with this regression line. From Figure 5, for T < -38°C, $D_{mean} \approx 50$ μm

appears reasonably representative for calculating $\alpha$ and $\beta$ for the cloud ice fraction. From Table 1 of EM2016, assuming synoptic cirrus clouds where -55°C < T < -40°C, $\alpha$ = 2.8755 (mks units) and $\beta$ = 2.5523. For cirrus clouds where T < -38°C, the sedimenting mass flux (i.e. the snow fraction) will be associated with $D_m$, which is about 3 times larger than $D_{mean}$ (Mitchell, 1990). Since our representative $D_{mean}$ is 50 μm, our representative $D_m$ is ~ 150 μm, which corresponds to $\beta$ = 2.2846 and $\alpha$ = 0.2350 (mks units). These values of $\alpha$ and $\beta$ are used to calculate $N_i$

and $N_s$ for cloud ice and snow in Eq. (24).

The so-called PSD y-intercept parameter $N_0$ is used in the MG2 scheme for calculating vapor deposition/sublimation (along with $\lambda$), as well as other processes. For $\nu = 0$, it is calculated as:





$N_0 = N \lambda.$ (26)

### 2.4 Ice particle growth processes

The above changes can affect ice particle growth/sublimation through water vapor diffusion and collection

processes. In this treatment, growth and sublimation through diffusion are affected through $\lambda$ and N, but collection

processes (i.e. snow self-aggregation, accretion of cloud ice by snow) that have an ice fall speed dependence are not

affected by these changes. Rather, the original MG2 ice fall speed parameters (*a* and *b*) are used for these collection

processes. Ice fall speeds based on the original MG2 fall speed parameters for cloud ice and snow agree with this

new treatment roughly within a factor of 2, and thus should be adequate for estimating removal rates of cloud ice by

snow and snowfall aggregation rates. The Bergeron process subroutine was not affected since this requires mixed

phase conditions (only ice exists in cirrus clouds).

For context, the accretion of cloud ice by snow in MG2 is expressed by a removal rate by snow, $R_{acc}$, times the IWC:

$$R_{acc} = \int_0^\infty V_s(D) \, A(D) \, E_{i,s} \, N(D) \, dD = \int_0^\infty a_s \, D^{b_s} \, (\pi/4) \, D^2 \, E_{i,s} \, N_{0,s} \, \exp(-\lambda D) \, dD = (\pi/4) \, a_s \, N_{0,s} \, E_{i,s} \, \Gamma(b_s + 3)/\lambda_s^{b_s+3} \quad (27)$$

where $V_s$ = snow fall speed, A(D) = projected area of snow particle, $E_{i,s}$ = collection efficiency of cloud ice by snow,

and N(D) is the PSD. Note that Eq. (27) assumes $V_s \gg V_i$, where $V_i$ = fall speed of cloud ice (this may

overestimate $R_{cc}$ for cirrus clouds in the modified MG2 scheme). In this modified MG2 scheme, $\lambda_i = \lambda_s = \lambda =$

$1/D_{mean}$, $N_{0,i} = N_i \lambda_i$, $N_{0,s} = N_s \lambda_s$, and standard MG2 values remain for $a_s$ and $b_s$, where subscripts i and s denotes the

cloud ice and snow fraction, respectively. Since IWC = $\rho_a q_i$, where $\rho_a$ = air density and $q_i$ = cloud ice mixing ratio,

the mass accretion rate is $R_{acc} \times$ IWC and the N accretion rate is $R_{acc}$ N.

### 2.5 Smoothing $D_e$ transitions

As mentioned, the CALIPSO $D_e$ retrievals were partitioned into 24 latitude and seasonally dependent categories

over land and 24 of such categories over ocean in the modified MG2 scheme. Sudden $D_e$ differences can sometimes

occur across category boundaries unless some kind of smoothing algorithm is applied. To smooth $D_e$ transitions

across categories, two latitudes were defined that were $\pm 5°$ latitude from the latitude being solved for, and the $D_e$

subroutine then yielded $D_e$ values for these two latitudes. We now have two $D_e$ values and two latitude values from

which to define a line equation, which is used to calculate the original $D_e$ (between these two latitudes). This

procedure prevents any sudden jumps in $D_e$ value from occurring and produces a $D_e$ gradient over $10°$ latitude.



### 2.6    Preparation of CALIPSO $D_e$ data and a MG2-CALIPSO subroutine for $D_e$

The $D_e$ subroutine in the modified MG2 scheme provides cirrus cloud $D_e$ values at 4 K temperature resolution for $168\ K \leq T \leq 268\ K$, but $D_e$ was not sampled across this entire temperature range.  The $D_e$ values beyond the minimum and maximum sampled $D_e$ take on the value of the minimum and maximum sampled $D_e$, respectively.

Appendix A contains a more detailed account describing how unsampled $D_e$ values were interpolated or extrapolated from the sampled $D_e$.  The $D_e$ subroutine (in Fortran90) used in MG2/WACCM6 that contains the LUTs giving $D_e$ as a function of T for each $D_e$ category (that are functions of season, latitude and ocean vs. land) is provided as a supplement to this manuscript as a possible resource to investigators who would like to pursue this type of research.  An explanation of the subroutine is also provided.

### 3    Experimental design

As mentioned, this study uses CALIPSO $D_e$ retrievals to constrain PSDs and ice fall speeds in WACCM6.  These retrievals apply to single layer cirrus clouds having an OD range from 0.3 to 3.0.  As explained in Sect. 6.3 of M2018, cirrus clouds in this OD range should dominate the overall cirrus cloud net radiative forcing, making their cloud properties relevant to climate modeling experiments that seek to understand cirrus cloud radiative effects.

In order to evaluate the radiative effect of hom relative to het, a second modified version of the MG2 scheme was produced that estimates cirrus cloud microphysical properties resulting from het.  This second version is the same as the first modified version described in Sect. 2 with the exception that the CALIPSO $D_e$ – T relationships obtained from the tropics ($\pm 30°$ latitude) are applied globally.  That is, for a given T, the $D_e$ for latitude zones 0 – 30 N and 0 – 30 S are averaged for each season, and the resulting LUTs for $\pm 30°$ latitude are applied to all six 30° latitude

zones.  This was done since retrieved N was lowest ($\sim$ 50 to 100 $L^{-1}$ based on the retrieval version most valid for the tropics; see M2018, Fig. 12b) in this $\pm 30°$ latitude zone relative to N at latitudes outside this zone, suggesting that the corresponding tropical $D_e$ would be the most representative of $D_e$ from cirrus clouds formed via het.  For this reason, the WACCM6 simulation corresponding to this second modified version of MG2 is referred to as HET. Although mechanisms/processes other than het may be responsible for N in the tropics (e.g. Lawson et al. 2015;

2017), similar low or minimal N are retrieved for regions outside the tropics where het likely dominates.  But such regions outside the tropics are not as spatially extensive as the tropics, making the tropics the most convenient region for developing "het" $D_e$ – T relationships.

In total, this modeling experiment consists of three 40-year simulations that are based on standard WACCM6, WACCM6 using the modified MG2 scheme as described in Sect. 2, and the HET version of WACCM6 as described

above.  Henceforth, these simulations will be referred to as WACCM, CALCAL (for calibrated with CALIPSO $D_e$) and HET, as shown in Table 1.  The "specified chemistry" version of WACCM6 (which runs faster than the



complete WACCM6) is used here since this study is not expected to be sensitive to changes in atmospheric
chemistry.  The default ice nucleation scheme of Liu and Penner (2005) is used in WACCM, where the pre-existing
ice option is switched on.  This option enhances the influence of het, reducing N by a factor of ~ 10 in the mid- to
high latitudes (Shi et al., 2015).  Differences between CALCAL and WACCM are intended to reveal deficiencies in

the treatment of ice nucleation in the MG2 scheme, while differences between CALCAL and HET are intended to
estimate the microphysical and radiative contribution of hom in natural cirrus clouds.  The simulation period is
1975-2014.  The horizontal resolution is 0.9° latitude × 1.25° longitude, and there are 70 vertical levels.
Climatological sea surface temperatures are used, which neutralize potential feedback effects from the ocean.  Each
simulation is "spun-up" to achieve balanced (stable) initial conditions before the simulation period commences.

As described in M2018, one has a selection of CALIPSO retrieval formulations to choose from; (1) based on
SPARTICUS in situ data with the smallest size-bin of the 2D-S probe [$N(D)_1$] included; (2) same but not including
$N(D)_1$; (3) based on TC4 in situ data with $N(D)_1$ included; (4) same but not including $N(D)_1$.  It was found in M2018
that formulations (1) and (2) yielded comparable agreement in the temperature dependence of $\beta_{eff}$ obtained from two
methods; direct measurement of $\beta_{eff}$ by CALIPSO IIR during SPARTICUS (considered most reliable) and

calculation of $\beta_{eff}$ from SPARTICUS PSD measurements.  Since the SPARTICUS field campaign was conducted in
the central United States (over or near the Rocky Mountains and often over low-lying plains), this suggests that
either retrieval formulation (1) or (2) may yield realistic retrievals outside the tropics, but formulation (1) yielded
superior agreement with in situ $D_e$ measurements during SPARTICUS.  The TC4 field campaign was conducted in
the tropics near Costa Rica, usually over ocean.  As described in M2018, formulation (4) produced the best

agreement in the temperature dependence of $\beta_{eff}$ obtained from the above two methods (i.e. direct measurement of
$\beta_{eff}$ by CALIPSO IIR during TC4 and calculation of $\beta_{eff}$ from TC4 PSD measurements).  In this modeling study, we
used formulation (1) outside the tropics and formulation (4) within the tropics.  Differences in $D_e$ between
formulations (1) and (4) are shown in Fig. 11 of M2018.

### 3.1   Seasonality of cirrus cloud properties

Although based only on formulation (1) of the CALIPSO retrieval (which produces the highest N), Figs. 6 and 7
show the seasonal dependence of global distributions of median $D_e$ and median N for 2008 and 2013, which is
represented in the LUTs regarding the former.  The seasonal dependence of N shown in Fig. 7 was also found by
Sourdeval et al. (2018) at mid-to-high latitudes for winter vs. summer.  Ice-supersaturated regions (ISSRs) are
requisite for cirrus cloud formation, and between the tropopause and ~ 100 hPa below that level and between 40 and

60° N. latitude, observations of RHi and ISSR frequency exhibit a clear seasonal dependence, being highest during
winter (DJF) and lowest during summer (Petzold et al., 2020).  Since higher RHi may coincide with more frequent
incidences of hom, these RHi and ISSR observations appear to support the findings in Figs. 6 and 7 (assuming





higher N is associated with more frequent incidences of hom). Additional evidence for this seasonality can be found in the cirrus cloud reflectance measurements of Zhao et al. (2020), where reflectance from cirrus clouds is highest during winter and lowest during summer at mid-to-high latitudes.

As described in M2018 and shown in Figs. 6 and 7, $D_e$ is smaller and N is higher over mountainous terrain. This is likely due to greater contributions from hom that occur in sustained updrafts from low frequency gravity waves induced by the mountains. This was also observed (and explained similarly) in the remote sensing study by Gryspeerdt et al. (2018). During winter, there is less mixing from deep convection, which may reduce INP concentrations in the upper troposphere, allowing the RHi threshold for hom to be more frequently realized.

## 4   Comparisons of CALIPSO retrievals with an in situ cirrus cloud climatology

Although CALIPSO retrievals of $D_e$, N and IWC are compared against corresponding in situ measurements from three field campaigns in M2018, a recent study by Krämer et al. (2020) has expanded the in situ cirrus cloud property database described in Krämer et al. (2009) by a factor of 5 to 10 (depending on cloud property). Here we convert CALIPSO retrievals of $D_e$ to the spherical volume radius ($R_v$) of the mean ice particle mass, IWC/N, and compare the temperature dependence of this "retrieved" $R_v$ with the in situ $R_v$ from the Krämer et al. (2020)

climatology. To facilitate comparisons between model-predicted N and in situ N, in Appendix B, N is calculated for exponential gamma PSD (Eqn. 6) based on in situ $R_v$ and IWC climatology as a function of T and compared with the in situ N reported in Krämer et al. (2020). This N is denoted as $N_{calc}$. This same method is applied in Appendix B to calculate N based on retrieval-derived $R_v$ and climatological in situ IWC. This N, denoted $N_{CAL}$, is thus the N consistent with retrieved $D_e$ and climatological IWC. Comparisons between in situ N, $N_{calc}$ and $N_{CAL}$ are shown in

Fig. B2 of Appendix B.

As stated in Krämer et al. (2020), the influence of wave cirrus clouds in the lee of Norwegian mountains was diminished in this new dataset, with most flights over lowlands and ocean (see Fig. 1 of that paper). As shown and discussed in Gryspeerdt et al. (2018), M2018 and Figs. 6-7, relatively high N and small $D_e$ are associated with mountainous topography, suggesting an enhanced contribution from hom; the opposite was observed over ocean and

lowlands. To compare the M2018 $D_e$ with in situ $R_v$ from Krämer et al. (2020), $D_e$ retrieved over ocean in the tropics (± 30 °latitude) is averaged with $D_e$ over ocean in the Northern Hemisphere midlatitudes (30 °N – 60 °N). Since the Krämer et al. (2020) data has no seasonal dependence, retrieved $D_e$ is averaged over all seasons. Figure 3 in Krämer et al. (2020) shows that very few $R_v$ measurements were made in the Arctic, and the few that were made are near 60 °N latitude, which is why retrieved $D_e$ is restricted to the tropics and midlatitudes.

The relationship between $R_v$ and $D_e$ can be evaluated numerically by incrementing $D_{mean}$ and calculating $D_e$ by inverting Eq. (1) and calculating $R_v$ from:





$$R_v = [3\ IWC/(4\ \pi\ \rho_i\ N)]^{1/3} = [3/(4\ \pi\ \rho_i)]^{1/3}\ (IWC/N)^{1/3}\ . \qquad (28)$$

This calculation assumed a gamma exponential PSD as defined by Eq. (6) with $\nu = 0$, with ice particle m and A
defined by Eqns. (2) and (3) and calculated using the EM2016 relationships described in Sect. 2.1. For a given
IWC, N was calculated via Eq. (24). $R_v$ is related $D_e$ in this way in Fig. 8, shown by the red diamond symbols. Also

shown in Fig. 8 are curve fits relating $R_v$ to $D_e$ based on Eqns. (28) and (12), where N, $A_{PSD}$ and IWC come from
2D-S (two-dimensional stereo) probe PSD measurements during the SPARTICUS and TC4 field campaigns
(described in M2018). These in situ calculations are shown by the blue squares in Fig. 8. The curve fit equations
are given in Appendix B. It was found that the $R_v$-$D_e$ relationship is sensitive to $\nu$, and the closer the curve fit is to
the red diamond symbols, the more the measured PSD conform to exponential PSD. Due to this sensitivity to PSD

shape, in situ measurements were used to define the $R_v$-$D_e$ relationship, using the TC4 curve fit when $D_e$ was
retrieved from the tropics and the SPARTICUS curve fit for the midlatitudes.

The results of this analysis are shown in Fig. 9 where $R_{ice}$ (same as $R_v$) is plotted against sampling temperature for
the in situ climatology (black curve) and the $D_e$-derived $R_v$ values (blue curves). The blue-dotted curve gives the
overall $R_v$ averaged over the tropical and midlatitude oceans. The dashed blue curve gives $R_v$ for the tropical oceans

only; recall that the HET simulation is based on the $D_e$ – T relationships from the tropics since this is where N was
consistently lowest and $D_e$ largest, and thus representative of cirrus formed through het. A potential explanation for
the relative agreement between the dashed-blue curve and the black curve is that primarily het-dominated cirrus
clouds were sampled in Krämer et al. (2020). The elevated portion of the black curve between 208 K and 190 K is
not observed in either of the two $D_e$-derived curves based on two years of CALIPSO retrievals. This may be due to

limited in situ sampling over this temperature range since the numerous retrievals here should capture the
temperature trend of $D_e$. The dotted blue curve is below the dashed blue curve because $D_e$ tends to be smaller at
midlatitudes relative to the tropics.

A more detailed analysis is given in Appendix B where N is calculated from the in situ median IWC and $R_v$ given in
Krämer et al. (2020) based on Eq. (6) and exponential PSD ($\nu = 0$). This calculated N is appropriate for comparing

with N predicted from the MG2 scheme that uses these PSD assumptions. In addition, using the comparison in Fig.
9 between 208 K and 235 K where the $R_v$ trend with temperature is similar between the black and blue-dotted
curves, it was found that on average, overall retrieved $R_v$ was ~ 80% of the median in situ $R_v$. It is natural to ask
what this retrieved $R_v$ implies in terms of a N – T climatology, and how this climatology compares with other N – T
climatological measurements. Using this percentage to estimate retrieved $R_v$ from in situ median $R_v$, along with the

30    in situ median IWC, N was calculated as a function of temperature. As shown in Fig. B2 of Appendix B, this
temperature dependence of N is similar to the N – T dependence reported in Fig. 4 of Gryspeerdt et al. (2018).



## 5   Modeling results and discussion

Figures 10 and 11 show global distributions for annual means regarding $D_e$ and N (the in-cloud values) at 250 hPa, respectively, as predicted by the CALCAL, WACCM and HET simulations. The 250 hPa level is associated with lower cirrus (near 235 K) in the tropics ($\pm$ 30° latitude) and mid-to-high level cirrus at high latitudes. Since $D_e$

generally decreases with T, this contributes to the decrease in $D_e$ at higher latitudes. But since the CALCAL model simulation is based on $D_e$ in Fig. 6 showing a poleward decrease in $D_e$ (averaged over all T < 235 K), this is also partly responsible for the poleward decrease in $D_e$ in Fig. 10 regarding CALCAL, which shows the lowest $D_e$ outside the tropics. Cirrus cloud N in Fig. 11 is relatively high in CALCAL and HET relative to WACCM, being highest in CALCAL, and thus inversely proportional to relative changes in $D_e$ shown in Fig. 10. However, N also

depends on IWC, and higher than climatological IWCs could also contribute to relatively high N. For example, Fig. B2 indicates that N based on climatological IWCs and retrieved $D_e$ should generally range between 50 and 400 $L^{-1}$. The frequent prediction of N > 300 $L^{-1}$ suggests that predicted IWCs are high relative to the in situ climatology. On the other hand, predicted N in CALCAL is broadly consistent with retrieved N in Fig. 7, which (as mentioned above), corresponds to relatively thick cirrus that are most relevant to radiation transfer. As predicted by Eq. (24), N

in the tropics is relatively high where the IWC tends to be higher, such as along the ITCZ, the western tropical Pacific Ocean, Polynesia, the Amazon Basin and the Congo. Possibly because the $D_e$ subroutine in the modified MG2 scheme is not sensitive to land topography, N in Fig. 11 is not relatively high over mountainous terrain like the retrieved N is in Fig. 7. In regards to Figs. 6 and 7, $D_e$ tends to be larger and N lower outside the tropics during the summer season. This behavior (not shown) was captured in the CALCAL simulation.

WACCM – CALCAL and HET – CALCAL differences are shown in Fig. 12 for the annual means of $D_e$ and N. Outside the tropics, WACCM – CALCAL $D_e$ differences are up to ~ 100 µm whereas HET – CALCAL $D_e$ differences are up to ~ 20 µm. Within the tropics, HET – CALCAL $D_e$ and N differences are near zero due to the design of this experiment ($D_e$ in the tropics in HET and CALCAL is the same). While WACCM N is almost always less than CALCAL N, over the tropical oceans these N differences are relatively small but still substantial. Outside

the tropics, especially along the storm tracks, CALCAL N can be up to ~ 150 $L^{-1}$ higher relative to HET N (presumably due to hom), and up to ~ 400 $L^{-1}$ higher relative to WACCM. These N differences are largest during the winter season (not shown).

### 5.1   Cloud micro- and macrophysical HET – CALCAL differences

Since this study is on homogeneous ice nucleation, the HET – CALCAL differences will be the focus henceforth.

One of the most surprising results of this study was the impact of cirrus clouds on the mixed phase cloud layer below them. The upper panel in Figure 13 shows the dependence of HET – CALCAL $D_e$ zonal mean differences on pressure and latitude. Since $D_e$ in the $\pm$ 30° latitude zone is treated the same in HET and CALCAL, differences are





near zero within this zone. All differences are positive (shown by reddish tones) and only the upper portion corresponds to cirrus clouds. For example, in Fig. 12, the 250 hPa HET – CALCAL $D_e$ differences are typically ~ 12 μm in the NH, but Fig. 13 shows much larger $D_e$ differences below this cirrus cloud layer that correspond to the zone of mixed phase clouds. This larger $D_e$ difference appears to result partially from differing fluxes of

sedimenting ice particles from overlying cirrus clouds and partially from differences in ice fall speeds. In the former, assuming sedimenting cirrus ice crystals contribute substantially to N in the mixed phase zone, N in this zone should be lower in the HET simulation, with less competition among ice particles for water vapor (i.e. RHi will be higher), with individual ice particles growing larger. In the latter, ventilation effects on diffusional growth (which can more than double stationary diffusional growth rates) are stronger at higher ice fall speeds, increasing $D_e$

in the mixed phase zone. Higher ice fall speeds also increase collision kernels for aggregation and riming, increasing those respective growth rates. The higher growth rates from all these processes further increase fall speeds, allowing $D_e$ to increase synergistically in the mixed phase zone.

A similar pattern is seen in the lower panel of Fig. 13, which shows HET – CALCAL zonal mean differences for the ice mass mixing ratio. This is always negative due to lower N in the HET simulation, which is due to a lower flux

of sedimenting cirrus ice particles and the higher fall speeds of larger ice particles. These differences often constitute a 10% decrease or more relative to CALCAL.

HET – CALCAL differences in cloud fraction are shown in Fig. 14 (upper panel). The cirrus cloud zone is characterized by a decrease in cloud fraction relative to CALCAL, due to higher fall speeds in HET. That is, the flux of ice from the base of a cirrus cloud strongly affects its lifetime and thus cloud fraction (e.g. Mitchell et al.,

2008). This enhanced export of ice from the cirrus zone lowers the RHi in the upper troposphere (lower panel), but this ice export increases the RHi in mixed phase zone, which increases cloud lifetime and thus cloud fraction there. The percent RHi changes in the mixed phase zone are generally greater than the changes in the cirrus cloud zone. This is also true for cloud fraction in the Northern Hemisphere (NH).

## 5.2    Understanding the CRE differences

The cloud radiative effect or CRE is defined as the sum of the top-of-atmosphere (TOA) net longwave (LW) and net shortwave (SW) cloud radiative effects, denoted LWCRE and SWCRE in Table 2. Also in Table 2 are the TOA net LW and SW fluxes for all sky conditions (that include clouds), denoted FLNT and FSNT. If these same TOA net fluxes for clear sky conditions (i.e. no clouds) are denoted as FLNTC and FSNTC, respectively, then LWCRE = FLNT – FLNTC and SWCRE = FSNT – FSNTC, and CRE = LWCRE + SWCRE. The performance of WACCM6

is documented in Gettelman et al. (2019), and as a check on the integrity of model output, Table 2 compares FLNT, FSNT, LWCRE and SWCRE from our 40-year WACCM simulation with the corresponding WACCM6 variables reported in Table 3 of Gettelman et al. (2019), as well as their observed values (also taken from this Table 3). Table





shows that the 40-year simulation values are very close to those reported in Gettelman et al. (2019), indicating these fluxes are accurate.

Table 3 shows the CRE differences regarding the WACCM, CALCAL and HET simulations. Although the MG2 scheme was only changed for $T \leq 235$ K, the CRE differences are substantial due to $D_e$ changes in the mixed phase zone (described above). The CRE for ice clouds depends strongly on the cloud optical thickness $\tau$ and the cloud fraction, where at solar wavelengths,

$$\tau = 3 \ IWP/(\rho_i \ D_e) \ , \tag{29}$$

where IWP is the ice water path and $\rho_i$ is the bulk density of ice (e.g. Mitchell, 2002). With generally larger $D_e$ and lower IWP in the HET simulation, $\tau$ tends to be lower in HET relative to CALCAL. If cloud fraction stays constant, this would result in lower SWCRE and LWCRE in HET. But the radiative effects resulting from changes (HET – CALCAL) in the upper- and mid-level cloud fraction adds complexity to the calculation of CRE. The combined effect is a more negative HET net CRE relative to CALCAL, as shown in Fig. 15 and Table 3. The HET – CALCAL CRE difference has a strong seasonal dependence, as shown in Fig. 15, where ΔCRE is as large as -5 W m$^{-2}$ over large regions at winter latitudes.

The reason the HET – CALCAL CRE difference is substantial at higher latitudes relates to the dependence of absorbed solar radiation and outgoing longwave radiation (OLR) on latitude and season. The annual average of these two fluxes is equal around $\pm$ 38 °latitude (Petty, 2006), with OLR dominating at higher latitudes. During winter, this OLR dominance increases and decreases during summer due primarily to the changing noontime solar zenith angle. Since cirrus clouds absorb LW radiation more efficiently than they reflect SW radiation, the cirrus CRE is determined primarily by OLR which dominates the radiation budget most at high latitudes during winter. At lower latitudes, SWCRE and LWCRE in cirrus clouds are comparable and tend to cancel so that the HET – CALCAL CRE is minimal. All this accounts for the latitude and seasonal dependence in Fig. 15.

While the HET – CALCAL CRE differences outside the tropics are considerable in Table 3 (-2.37 W m$^{-2}$ for the N.H. and -2.55 W m$^{-2}$ for the S.H.), the global mean CRE difference is considerably less (-1.16 W m$^{-2}$). This is because in the $\pm$ 30 °latitude zone, $D_e$ – T relationships were identical in HET and CALCAL. But even if these $D_e$ – T relationships differed, this latitude zone would contribute little to the global mean CRE difference since, as noted above, the cirrus cloud SWCRE and LWCRE tend to cancel in the tropics (Storelvmo and Herger, 2014).

## 5.3  TOA net radiative fluxes

In the final analysis, what matters most to climate are the differences in TOA net flux (FSNT - FLNT) between simulations; these are shown in Table 4. Although these net flux differences are mostly due to CRE differences,





they are also partly due to differences in absorption/emission by water vapor that affect FLNT; we call this the relative humidity or RH radiative effect. As shown in Fig. 14, HET has relatively less cloud fraction and RH in the upper troposphere (UT) relative to CALCAL, and relatively higher cloud fraction and RH at mid-levels relative to CALCAL. However, the mid-level changes are greater than the UT changes (both in magnitude and vertical extent),

which acts to reduce FLNTC more in HET. This appears to be the main cause of the difference between the CRE changes in Table 3 and the TOA net flux changes in Table 4. HET – CALCAL differences in TOA net flux are shown in Fig. 16 based on annual means and for winter verses summer. As with the CRE differences, there is a strong seasonal dependence. The RH radiative effect is strongest in the Polar Regions, especially during the N. H. summer. Table 5 shows TOA net flux differences outside the tropics, similar to Table 4, except that the summer

months (JJA in the N.H. and DJF in the S.H.) are omitted from the averaging. These differences in Table 5 are greater than in Table 4 due to three factors that tend to be greater during non-summer months at these latitudes: hom contributions to cirrus formation, the dominance of OLR (relative to incoming SW) radiation, and a diminished (relative to summer) RH radiative effect that counteracts the CRE.

In addition to RH, there are other factors affecting the difference between simulations regarding TOA net flux and

net CRE. The clear sky fluxes are also sensitive to aerosols, and to other radiatively active gases. Since this study uses the "specified chemistry" version of WACCM6, it is less likely that ozone and other greenhouse gas concentrations are significantly changing between simulations. Nonetheless, changing clouds will change actinic fluxes and photolysis rates, and thus reactive gas concentrations, which could also affect the clear sky fluxes. Aerosol concentrations can also be changed by modest circulation changes that change scavenging rates.

In Tables 3 and 4, since WACCM $D_e$ > HET $D_e$ > CALCAL $D_e$, one might expect the WACCM – CALCAL radiation differences to exceed the HET – CALCAL differences. But these differences also depend on the cloud IWC, and IWCs could be higher in WACCM at least partly because WACCM ice fall speeds tend to be lower than those in HET and CALCAL; see Fig. 4. Note that ice fall speeds can strongly impact IWCs (Mitchell et al., 2008).

## 6    Conclusions

As discussed in M2018 and Sect. 1, and also in other recent studies (Zhao et al., 2018; Sourdeval et al., 2018; Gryspeerdt et al., 2018), homogeneous ice nucleation (hom) contributes substantially to the microphysical and radiative properties of cirrus clouds, depending on cooling rate, latitude, season and topography. Explanations for why this happens are offered in these studies and in Sect. 1, which identifies two fundamental reasons: (1) higher cooling rates (updrafts) in mountain-induced waves and (2) the hom RHi threshold is more easily reached when INP

concentrations are relatively low. In addition, INP concentrations at cirrus levels probably depend on deep convection, which occurs much more often during summer. Also, pre-existing ice advected into anvil cirrus competes effectively for water vapor, suppressing RHi and reducing the frequency of hom. These may be the



reasons that, outside the tropics, $D_e$ tends to be largest and N lowest during the summer season as shown in Figs. 6 and 7. The seasonal behavior of $D_e$ and N shown in Fig. 6 and 7 is probably not sufficiently realized in climate models. This study attempts to estimate the radiative impact of this seasonal behavior.

From the HET – CALCAL differences in Table 3, one can infer that the CRE is sensitive to hom, and that the hom
CRE effect is large enough to affect climate. For example, it is possible that climate sensitivity depends on INP concentration, where climate sensitivity is the global mean change in surface temperature resulting from a doubling of $CO_2$ at radiative equilibrium. If an increase in INP concentration were to neutralize the hom CRE effect, then it is possible that this could significantly decrease the global mean surface temperature. However, to adequately test this hypothesis, fully coupled simulations would be needed (i.e. these are atmosphere global climate model simulations).

Many attempts to estimate the radiative contribution of hom (relative to het) were addressing the efficacy of the climate intervention method often referred to as cirrus cloud thinning or CCT (e.g. Storelvmo et al., 2013; Muri et al., 2014; Storelvmo and Herger, 2014; Storelvmo et al., 2014; Crook et al., 2015; Kristjansson et al., 2015; Penner et al., 2015; Gasparini and Lohmann, 2016; Gasparini et al., 2017; Muri et al., 2018; Gruber et al., 2019; Gasparini et al., 2020). These were mostly global climate modeling studies (Gruber et al., 2019, was a regional modeling
study) that differed in their treatment of hom and het as well as other processes affecting the frequency of hom. Not surprisingly, given the multitude of variables affecting hom and het, the impact of converting hom to het through seeding the UT with efficient ice nuclei ranged from a global annual mean CRE difference of - 2.2 W m$^{-2}$ (Storelvmo et al., 2014) to essentially no impact (e.g. Penner et al., 2015; Gasparini and Lohmann, 2016). This study estimates a global annual mean HET – CALCAL CRE difference of -1.16 W m$^{-2}$. However, outside the $\pm$ 30
°latitude zone, the annual mean CRE difference is -2.37 W m$^{-2}$ in the N.H. and -2.55 W m$^{-2}$ in the S.H., and during non-summer months when CCT is effective, the TOA HET – CALCAL net flux difference is -2.4 W m$^{-2}$ in both hemispheres. Based on this study, CCT may have a significant cooling impact at mid-to-high latitudes, but to estimate the CCT impact on surface temperatures, coupled simulations are needed that consider feedback effects from the oceans, cryosphere and land components of the earth-system.

In addition to its relevance to CCT, this study identifies an apparent deficiency in climate modeling since the indicated radiative contribution of hom and associated heating rates at mid-to-high latitudes are currently missing in climate models that explicitly treat cirrus cloud microphysics (e.g. Gasparini and Lohmann, 2016). This can have an impact on geopotential height patterns, the storm track and precipitation patterns (Li et al., 2015; 2016; 2017). Moreover, the hom radiative effects are most pronounced at mid-to-high latitudes during winter, which may affect
Arctic amplification (Holland et al., 2003) and climate tipping point phenomena (Holland et al., 2006; Notz et al., 2009). For a discussion on a possible linkage between high latitude cirrus clouds and midlatitude weather, see Sect. 6.5 of M2018.



Finally, this study indicates that changes in cirrus cloud $D_e$ and N produce large consequences in the underlying mixed phase clouds and relative humidity field as described in Sect. 5.1. These consequences contribute strongly to the CRE and the TOA net radiative fluxes. More research on the microphysical processes producing these changes in mixed phase cloud properties may be needed to corroborate these findings. Note that changes to the MG2 scheme

5    were only made for cirrus clouds where $T \leq 235$ K.

As mentioned, the atmospheric simulations in this study were not coupled with other components of the earth-system, such as the oceans, land and cryosphere, thus preventing important climate feedbacks from occurring that affect the general circulation and surface temperatures. Future research should conduct such coupled simulations to understand how potential changes in INP concentration could affect the Earth's climate. Moreover, there are four

10   versions of the CALIPSO retrieval for $D_e$ and N, where N can vary by about a factor of two, depending on which version is used (M2018). This modeling study could be repeated using different retrieval versions to characterize the uncertainty associated with the results reported here.



## Appendix A: Methodology for preparing $D_e$ – T LUTs for the modified MG2 scheme

A new subroutine in the modified MG2 scheme relates a retrieved median $D_e$ to a given temperature (T), latitude, season and land fraction using look-up-tables (LUTs). Each LUT consists of an array of $D_e$ values corresponding to T at 4 K intervals, where the T range is from 168 – 268 K. However, the T range of retrieved $D_e$ is from 188 – 235
K since only cirrus clouds are targeted by the retrieval. Interpolation is used within the cirrus T range to calculate $D_e$ when T lies within a prescribed 4 K interval. The following guidelines were adopted for assigning $D_e$ values in LUTs for T having no corresponding CALIPSO satellite $D_e$ measurement or for identifying/replacing unrealistic $D_e$ values. These selection rules assume $D_e$ decreases with decreasing T; $D_e$ violating this trend are usually based on a relatively small number of samples. See Fig. 11 in M2018 for graphical displays of $D_e$ – T data used to create
LUTs.

1) If retrieved median $D_e$ corresponding to the highest T is lower than the adjacent $D_e$ (at lower T), set $D_e$ for the highest T equal to the adjacent $D_e$ (producing two equal-valued $D_e$ for the two warmest T points).

2) If retrieved median $D_e$ for the lowest T is higher than the adjacent $D_e$ (at higher T), set $D_e$ for the lowest T equal to the adjacent $D_e$ (producing two equal-valued $D_e$ for the two coldest T points).

3) If median $D_e$ for the two lowest T appear non-physical (i.e. they usually follow a negative slope and have $D_e$ values larger than $D_e$ for the 3rd lowest T), then assign them the $D_e$ value of the 3rd lowest T if the number of samples corresponding to these two anomalous $D_e$ is < ~ 100. Number of $D_e$ values affected = 4.

4) When number of samples N < 10 and the median $D_e$ is anomalous relative to adjacent values, interpolate between two adjacent $D_e$ values to obtain the intermediate value. Number of $D_e$ values affected = 3.

5) If median $D_e$ for the highest T is more than 20 μm higher than the adjacent $D_e$-T point, and N < 10, then set the $D_e$ for the highest T as equal to the adjacent $D_e$ (producing two equal-valued $D_e$ for the two warmest T points). Number of $D_e$ values affected = 3.

## Appendix B: Calculating N from retrieval-derived $R_v$ and in situ climatology IWC

This appendix first provides the curve fit equations used in Fig. 8 to convert retrieved $D_e$ to $R_v$, and then proceeds to describe a methodology for calculating N from $R_v$ and IWC. Regarding the latter, first the in situ $R_v$ is used, followed by $R_v$ estimated from the median $D_e$ retrievals averaged from the tropics and midlatitudes. The purpose of this exercise was to first calculate in situ climatological values for N (denoted $N_{calc}$) that conform to assumptions in





climate models, such as an analytical exponential gamma function which is the case here. The second purpose was to calculate N based on retrieved $D_e$ and in situ climatological IWCs that should provide climatological values for N that are consistent with these two climatological properties ($D_e$ and IWC). This retrieval-derived N, denoted $N_{CAL}$, can then be related to other in situ and satellite climatologies of N.

**B1   Curve fits used in Figure 8**

As mentioned, $D_e$ retrievals at midlatitudes utilize the SPARTICUS unmodified PSD version of this retrieval primarily because this version yielded the best agreement between retrieved and in situ $D_e$ in retrieval validation studies (M2018). In the tropics, $D_e$ retrievals utilize the TC4 $N(D)_1 = 0$ version of this retrieval (where $N(D)_1$ denotes the number concentration in the smallest size bin of the 2D-S probe) because the CALIPSO IIR

measurement of the temperature dependence of $\beta_{eff}$ (the 10.6 to 12.05 µm effective absorption optical depth ratio) matched $\beta_{eff}$ derived from in situ PSD measurements the best when $N(D)_1 = 0$. These same microphysical assumptions were adhered to (unmodified PSD for SPARTICUS and $N(D)_1 = 0$ for TC4) when producing the curve fits shown below. The SPARTICUS curve fit shown in Fig. 8 is a 6-order polynomial function for $D_e \geq 45$ µm and a linear extrapolation for $D_e < 45$ um. For $D_e \geq 45$ um,

$$R_v = a_0 + a_1 D_e + a_2 D_e{}^2 + a_3 D_e{}^3 + a_4 D_e{}^4 + a_5 D_e{}^5 + a_6 D_e{}^6, \qquad (B1)$$

with $a_0 = 33.2197$, $a_1 = -4.33987$, $a_2 = 0.261068$, $a_3 = -0.00712459$, $a_4 = 9.93114e-05$, $a_5 = -6.65497e-07$ and $a_6 = 1.69976e-09$, where units are microns. For $D_e < 45$ um,

$$R_v = 0.310553 (D_e - 45) + 15.9104, \qquad (B2)$$

where again units are microns. The linear extrapolation is well adapted in the range of $D_e$ that we observe.

For the TC4 curve fit shown in Fig. 8,

$$R_v = a_0 + a_1 D_e + a_2 D_e{}^2 + a_3 D_e{}^3, \qquad (B3)$$

where $a_0 = 5.58658$, $a_1 = 0.290955$, $a_2 = -0.00266899$ and $a_3 = 5.51555e-05$ and units are microns. For $D_e < 18$ µm,

$$R_v = 0.5598 D_e. \qquad (B4)$$

**B2   Methodology for calculating N from in situ $R_v$ and IWC**

The same curve-fitting methodology described above was applied here in relating the dependent variable $D_e$ to the independent variable $R_v$, where $R_v$ comes from the in situ climatology of Krämer et al. (2020). In Sect. B1 the goal was to obtain $R_v$ from retrieved $D_e$ for the purpose of comparison with in situ $R_v$ in Fig. 9. In this section, the goal is





to obtain $D_e$ from in situ $R_v$ (or an estimate of retrieved $R_v$ from in situ $R_v$), and then estimate N from this $D_e$ and the in situ climatology IWC. The $D_e$ – $R_v$ curve fit for the midlatitudes using SPARTICUS data is shown in Figure B1 (black curve) for T < 235 K and is described mathematically with micron units as

$$D_e = -11.6202 + 4.09910\ R_v + 0.00767864\ R_v^2 - 0.00371768\ R_v^3 + 8.72769e\text{-}05\ R_v^4 - 8.01817e\text{-}07\ R_v^5$$

$$+ 2.64619e\text{-}09\ R_v^6. \tag{B5}$$

For $R_v$ < 5 µm, $D_e$ is given by

$$D_e = 1.7309\ R_v. \tag{B6}$$

The $D_e$ – $R_v$ curve fit for the tropics (± 30 °latitude) using TC4 data is shown in Figure B1 (black curve) for T < 235 K and is described mathematically with micron units as

$$D_e = -25.4617 + 5.00679\ R_v - 0.0709877\ R_v^2 + 0.000354202\ R_v^3. \tag{B7}$$

For $R_v$ < 10 µm,

$$D_e = 1.7862\ R_v. \tag{B8}$$

These relationships account for PSD shape effects in anvil cirrus (sampled during TC4) and in synoptic (non-anvil) cirrus (sampled during SPARTICUS), which are assumed to be representative of PSD shape effects encountered in

tropical and midlatitude cirrus analyzed in Krämer et al. (2020), respectively. Again, the red diamonds in Fig. B1 were derived from (1) and (28) using the m-D and A-D relationships in EM2016 for synoptic cirrus clouds between -40 °C and -55 °C (described in Sect. 2.1), and assuming exponential gamma PSD. Differences between these $D_e$-$R_v$ values and the curve fits are believed to result from PSD shape differences that can have a large impact on N.

To obtain the final $D_e$ that represents in situ sampling in both the tropics and midlatitudes, these two $D_e$ values, one

from the SPARTICUS relationship and one from the TC4 relationship, are averaged together. Equation (7) is then applied, which assumes exponential gamma PSD, to obtain $D_{mean}$. For exponential PSD, the slope parameter $\lambda = 1/D_{mean}$. From $\lambda$ and the median in situ IWC reported in both Krämer et al. (2009) and Krämer et al. (2020), N (denoted $N_{calc}$) is calculated from this in situ climatology that conforms to exponential gamma PSD. The temperature dependence of $N_{calc}$ and in situ N reported in Krämer et al. (2020) are compared in Fig. B2. There is

fairly good agreement between 244 K and 217 K, after which $N_{calc}$ exceeds in situ N to varying degrees (up to a factor of 8). It is difficult to determine the cause of this variance, but since in situ $R_v$ and IWC are conserved, it may relate to changes in PSD shape at the smallest ice particle sizes. For models that parameterize ice PSD using exponential gamma distributions, from a radiation purview, model-predicted climatological N comparisons against $N_{calc}$ may be more meaningful than using the raw in situ N measurements.

**B3   Methodology for calculating N from retrieval-derived $R_v$ and in situ IWC**

Over the temperature range from 235 K to 208 K in Fig. 9, the trend is similar regarding the in situ and retrieval based estimates of $R_v$. Thereafter, at lower T, there is an increase for in situ $R_v$ that is not observed in retrieval-based $R_v$ where the sampling statistics are much greater relative to in situ measurements. Since this tendency change for in situ N ~ 208 K is not found in the retrievals, it is argued that the N comparison between 208 and 235 K is

more reliable for evaluating the mean difference in $R_v$ between in situ and retrieval-based $R_v$. The ratio of the sum of retrieval-based to in situ $R_v$ values between 208 and 235 K is 0.7945 (where retrieved $R_v$ is the mean of tropical and midlatitude values). That is, retrieval-based $R_v$ ≈ 0.8 in situ $R_v$. Using this ratio, retrieval-based $R_v$ was estimated from in situ $R_v$ over the entire temperature range shown in Fig. 9. The rest of the procedure is the same as in Sect. B2, where $D_e$ is estimated from retrieval-based $R_v$, and $D_{mean}$ and $\lambda$ are calculated from $D_e$ via Eq. (7). From

$\lambda$ and the in situ IWC, N (denoted $N_{CAL}$ for CALIPSO) is calculated. The temperature dependence of $N_{CAL}$ is plotted





along with $N_{calc}$ and N in situ in Fig. B2. Since $N_{CAL}$ employs the same assumptions as $N_{calc}$ (excepting $R_v$), a comparison between $N_{CAL}$ and $N_{calc}$ is most appropriate. Between 244 K and 218 K, $N_{CAL} > N_{calc}$ by ~ 50% to 100%. For lower temperatures, $N_{CAL}$ can be up to a factor of 3 (i.e. 200%) higher than $N_{calc}$. Since $D_e$ is used to constrain WACCM6, this provides a more realistic appraisal in terms of N regarding the differences between our
CALIPSO $D_e$ retrievals and the in situ $R_v$ measurements in Krämer et al. (2020).

It should be appreciated that N is a difficult ice cloud property to measure using probes on aircraft and can easily vary by a factor of 2 or more depending on the instrument and data processing (e.g. Korolev et al., 2011; Lawson, 2011; Korolev and Field, 2015). Moreover, N retrievals using the DARDAR method (Sourdeval et al., 2018; Gryspeerdt et al., 2018) can vary by a factor of ~ 1.7 depending on whether the small ice crystal cut-off size used for
the assumed PSD shape is 5 µm or 1 µm (M2018). The temperature dependence of $N_{CAL}$ in Fig. B2 agrees well with the temperature-variation of retrieved N in Fig. 4 of Gryspeerdt et al. (2018) which uses 5 µm as the PSD cut-off size. The retrieval method used here does not assume any small ice crystal cut-off size, it considers all detectable particles, and the method is very sensitive to small ice crystals (M2018).

The N values shown in Fig. 7 are based on the CALIPSO retrieval version that yields the highest N estimates (i.e.
the SPARTICUS unmodified PSD version), and N varies by about a factor of 2 depending on which version is used. Therefore, these N estimates may be high by a factor of 2 if the version producing the lowest N estimate is most realistic. Moreover, the retrieval applies only to relatively thick cirrus clouds having optical depths between 0.3 and 3.0. These thicker cirrus may be associated with higher updrafts where the hom contribution is larger, resulting in higher N. On the other hand, such cirrus have the strongest influence on the earth's radiation budget and are thus
most representative for climate modeling purposes (M2018).





*Code availability*. The new code developed for this project is provided as a supplement (a subroutine for replacing model predicted $D_e$ with CALIPSO retrieved $D_e$ as a function of temperature, latitude, season and land fraction). Other new code implemented is described mathematically by the equations relating $D_e$ to PSD slope and $D_e$ to the number- and mass-weighted ice fall speeds in Sect. 2. Values of $\alpha$ and $\beta$ for calculating $N_i$, $N_s$, $N_{0i}$ and $N_{0s}$ in cirrus

clouds ($T < 235$ K) are given in Sect. 2.3.

*Data availability*. Access to the CALIPSO satellite data used here is described under "data availability" in M2018 and it is also available as code in the subroutine supplement described above.

*Author contributions*. DM conceived and directed the study and developed/implemented new code in the MG2 scheme; JM implemented new code, executed the simulations and processed model output; AG contributed the maps of $D_e$ and N retrievals and the conversion of $D_e$ to $R_v$; YT processed model output and developed some code; MK supplied the climatology of in situ cirrus cloud properties; FH participated in the debugging process.

*Competing interests*. The authors declare that they have no conflict of interest.

*Acknowledgements*. This research was supported by NASA grant NNX16AM11G and the NASA CALIPSO project. Fruitful discussions with Dr. Phil Rasch regarding these results are acknowledged and appreciated. The use of WACCM6 for this study was recommended by Dr. Julio Bacmeister and the forty-year simulation period was

suggested by Dr. Andrew Gettelman (to evaluate sudden stratospheric warming event frequencies); we thank both of them for their insights and guidance. CALIPSO products are available at the Atmospheric Science Data Center of the NASA Langley Research Center and at the AERIS/ICARE Data and Services Center in Lille (France). Resources supporting this work were provided by the NASA High-End Computing (HEC) Program through the NASA Center for Climate Simulation (NCCS) at Goddard Space Flight Center.

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



Table 1.  WACCM6 Cases Discussed in the Text

| Case Name | Description |
|---|---|
| WACCM | standard version of WACCM6 without modifications |
| CALCAL | WACCM6 with modified MG2 scheme based on CALIPSO $D_e$ retrievals |
| HET | WACCM6 with modified MG2 scheme based on CALIPSO $D_e$ retrievals from tropics only |

Table 2.  Global annual mean radiative flux comparisons (W m$^{-2}$) between our standard WACCM6 simulation, published WACCM6 results (Gettelman et al., 2019) and satellite observations from CERES EBAF 2.4 (Loeb et al., 2009).  FLNT and FSNT are top-of-model net longwave and shortwave fluxes for all sky conditions, respectively, while LWCRE and SWCRE are top-of-model net longwave and shortwave cloud radiative effects (FLNT & FSNT minus corresponding clear sky net fluxes), respectively.

| Variable | WACCM6 (40-year run) | Published WACCM6 results | Observations |
|---|---|---|---|
| FLNT | 237.7 | 237.4 | 239.7 |
| FSNT | 240.9 | 241.0 | 240.5 |
| LWCRE | 24.4 | 24.6 | 26.1 |
| SWCRE | -48.0 | -48.4 | -47.1 |

Table 3.  Differences in cloud radiative effect (CRE) for the simulations of this study for the global annual mean and for annual means outside the tropics, where NH = Northern Hemisphere (30N – 90N) and SH = Southern Hemisphere (30S – 90S).  Units are W m$^{-2}$.

| | WACCM – CALCAL | WACCM – HET | HET – CALCAL |
|---|---|---|---|
| Global | -0.132 | 1.03 | -1.16 |
| NH | -1.63 | 0.744 | -2.37 |
| SH | -2.59 | -0.0437 | -2.55 |

Table 4.  Differences in total net forcing at TOM (top of model) for the simulations of this study for the global annual mean and for annual means outside the tropics, where NH = Northern Hemisphere (30N – 90N) and SH = Southern Hemisphere (30S – 90S).  Units are W m$^{-2}$.

| | WACCM – CALCAL | WACCM – HET | HET – CALCAL |
|---|---|---|---|
| Global | 0.188 | 1.12 | -0.929 |
| NH | -1.12 | 0.654 | -1.77 |
| SH | -1.94 | 0.0887 | -2.03 |





Table 5. Differences in total net forcing at TOM (top of model) for means excluding summer in W m$^{-2}$, where NH = Northern Hemisphere (30N – 90N) and SH = Southern Hemisphere (30S – 90S). Excluding summer means that JJA is omitted for NH and DJF is omitted for the SH.

|  | WACCM – CALCAL | WACCM – HET | HET – CALCAL |
|---|---|---|---|
| NH | -1.56 | 0.864 | -2.43 |
| SH | -2.11 | 0.322 | -2.43 |



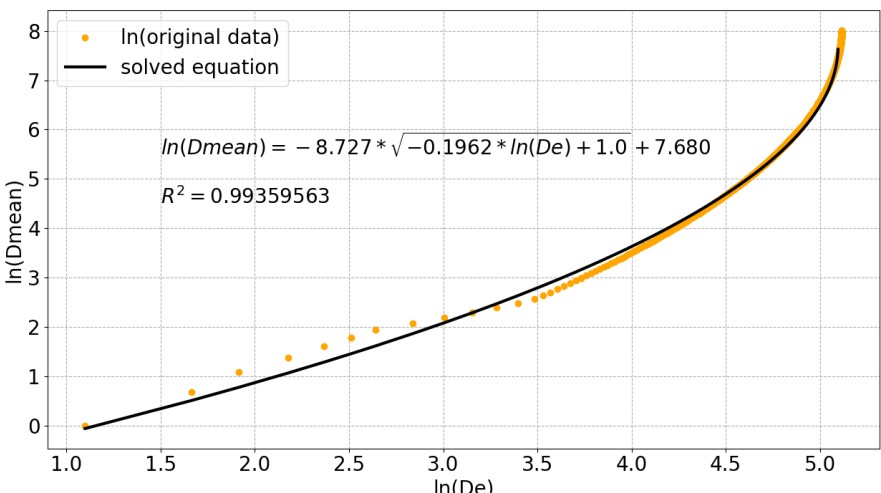

Figure 1. Relationship between $D_{mean}$ and $D_e$ (in µm) as predicted by Eq. 3 and the size (D) dependent m-D and A-D coefficients (α, β, γ and δ) as described in EM2016. The series of orange points were generated by the iterative
5    solution, and the dark curve was generated by the curve-fit equation shown.

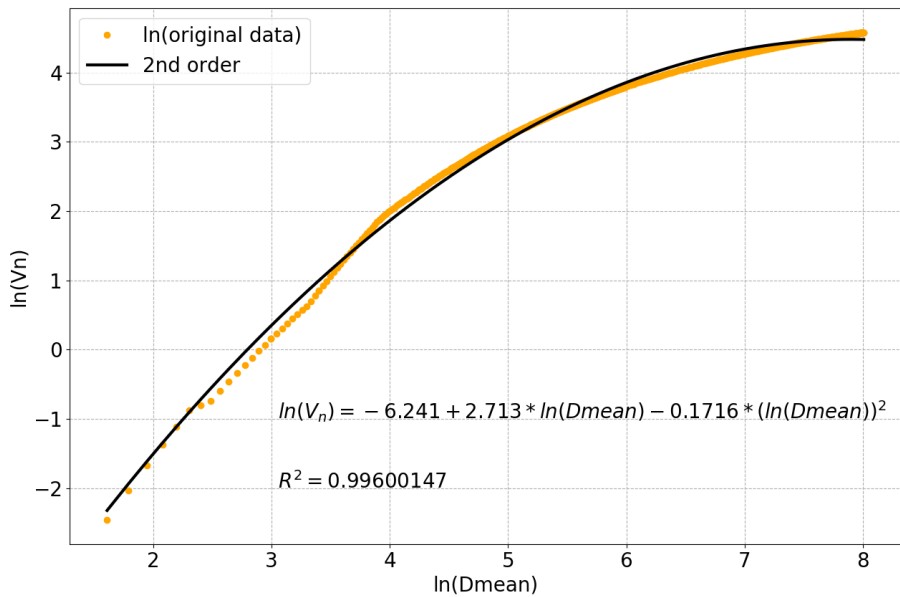

Figure 2. Comparison between generated $D_{mean}$-$V_N$ values (in µm and cm s$^{-1}$) based on the above methodology, and the corresponding curve-fit for use in the MG2 scheme.



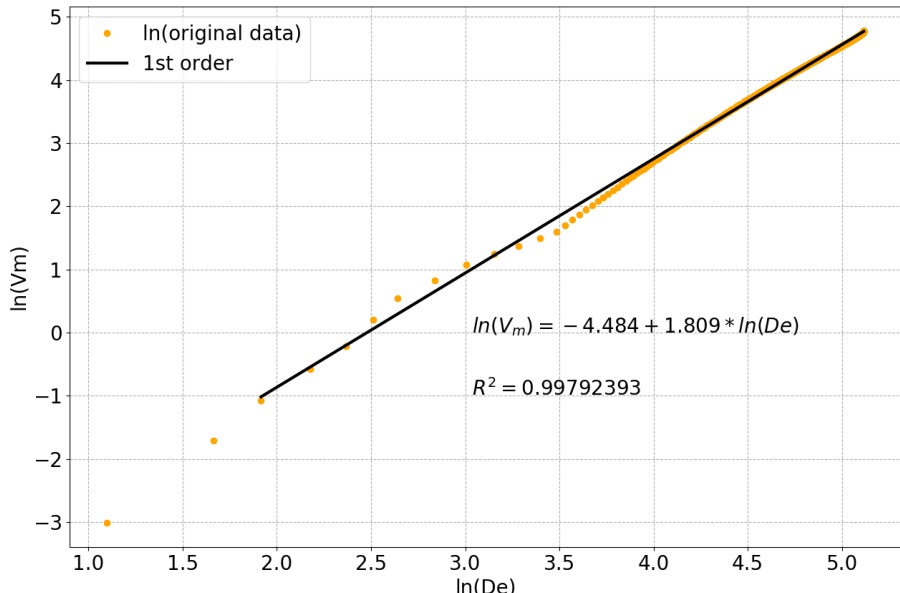

Figure 3. Comparison between generated $D_e$-$V_m$ values (in µm and cm s$^{-1}$) based on the above methodology, and
5    the corresponding linear-fit for use in the MG2 scheme.

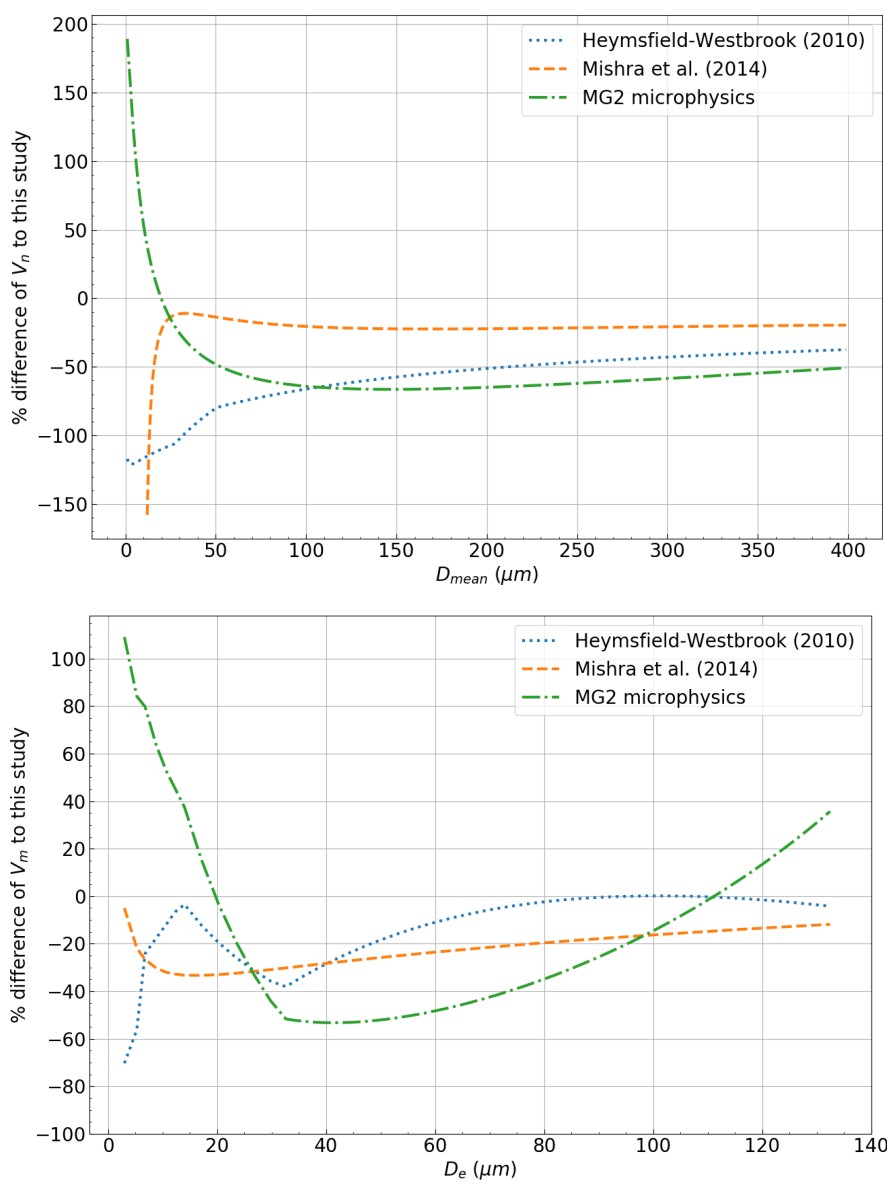

Figure 4. Comparisons of three ice fall speed schemes (including the MG2 scheme) against the scheme developed for this study. Both number- and mass-weighted fall speeds, $V_N$ and $V_m$ (top and lower panels, respectively), are calculated in each scheme. The closer to the zero line, the closer the agreement is.

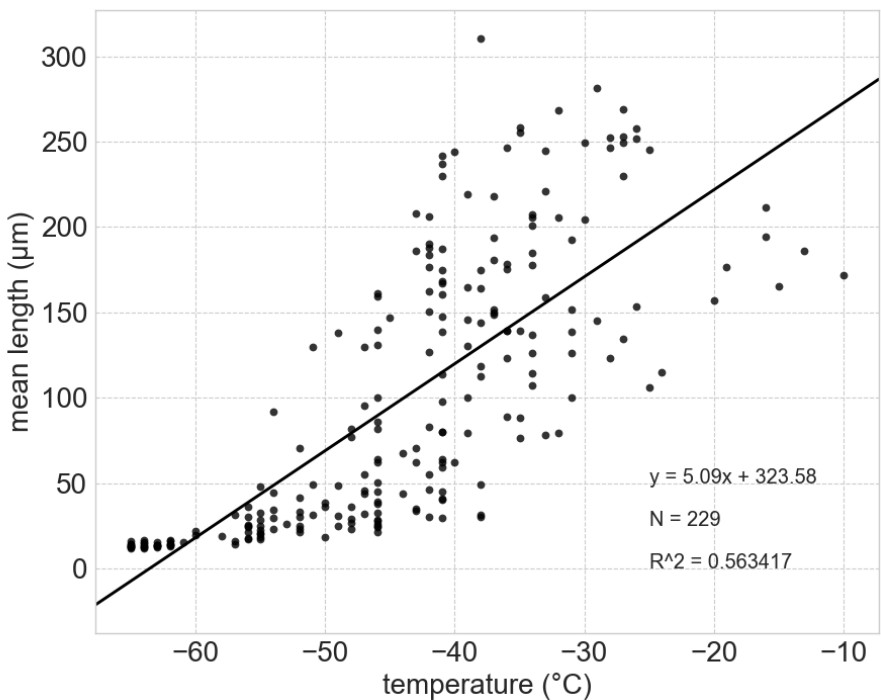

Figure 5. PSD mean maximum dimension ($D_{mean}$) related to sampling temperature, based on SPARTICUS data for synoptic (non-anvil) cirrus clouds. The regression line estimates $D_{mean}$ within a factor of 3, based on the spread for any given temperature. This should be sufficiently adequate for estimating the constants used in the mass and area power laws. Regarding the range of $D_{mean}$, $D_{mean}(-60°C) \approx 18$ μm and $D_{mean}(0°C) \approx 324$ μm. This dataset is based on unmodified 2DS data (i.e. the smallest 2DS size bin is included). Evidence of homogeneous ice nucleation is apparent for T < -40°C.



Figure 6. The seasonal dependence of median $D_e$ (color legend at center is in μm) for temperature $T \leq 235$ K for the years 2008 and 2013, based on the first formulation (SPARTICUS in situ data with $N(D)_1$ included) of the CALIPSO retrieval.

Figure 7. The seasonal dependence of median N (color legend at center is in $L^{-1}$) for $T \leq 235$ K for the years 2008 and 2013, based on the first formulation (SPARTICUS in situ data with $N(D)_1$ included) of the CALIPSO retrieval.

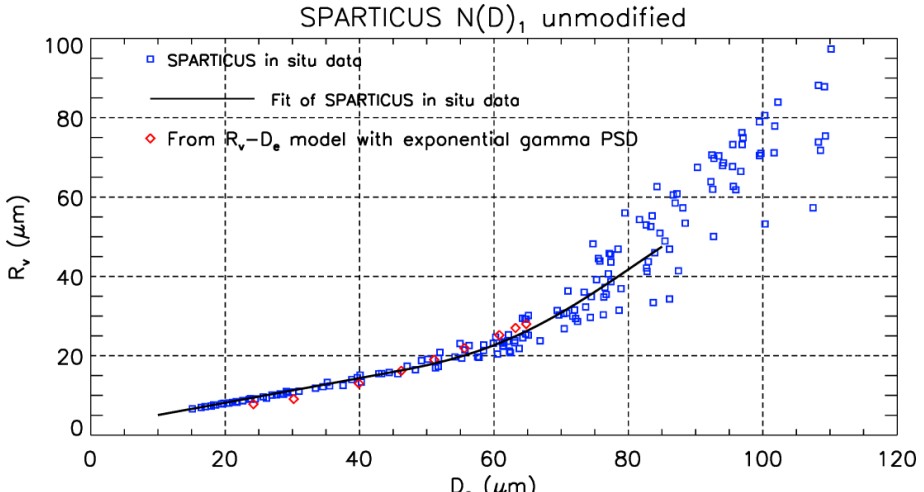

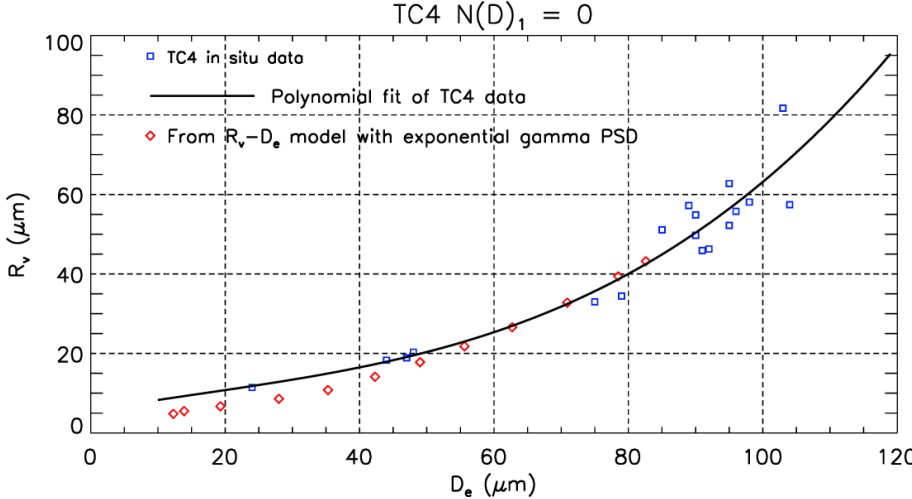

Figure 8. In situ measurements of N, $A_{PSD}$ and IWC were calculated from 2D-S probe PSD measurements (or estimates regarding IWC) taken during the SPARTICUS and TC4 field campaigns. These were used in Eqns. (12) and (28) to calculate $D_e$ and $R_v$ for each PSD, shown by the blue squares. The red diamonds were determined theoretically for exponential PSD (see text for details). The curve fits (see Appendix B) were used to convert
10   retrieved $D_e$ to $R_v$, where the TC4 fit is applied to tropical $D_e$ and the SPARTICUS fit to midlatitude $D_e$.

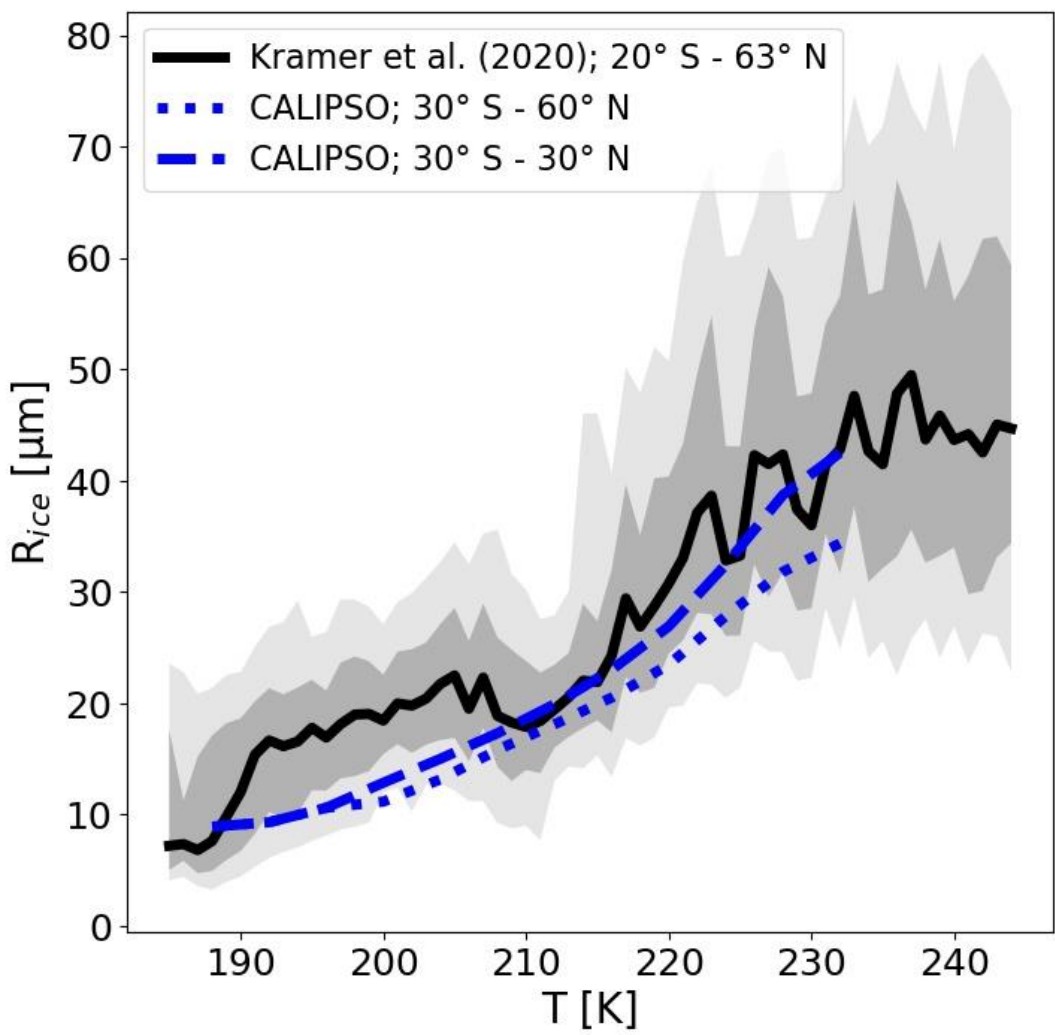

Figure 9. Comparison of the temperature-dependence of retrieval derived $R_v$ (i.e. $R_{ice}$; blue curves) with median $R_v$ from the in situ climatology of Krämer et al. (2020) shown by the black curve. The grey envelope gives the 25 and 75 percentiles while the light grey gives the 10 and 90 percentiles for the in situ $R_v$. The dashed blue curve is for the tropics only (± 30 °latitude) while the dotted blue curve is averaged over the midlatitudes and tropics.





Figure 10. From top-to-bottom, annual mean CALCAL, HET, and WACCM6 simulation results for in-cloud $D_e$ at 250 hPa in microns.



Figure 11. From top-to-bottom, annual mean CALCAL, HET and WACCM6 simulation results for in-cloud N at 250 hPa in L$^{-1}$.





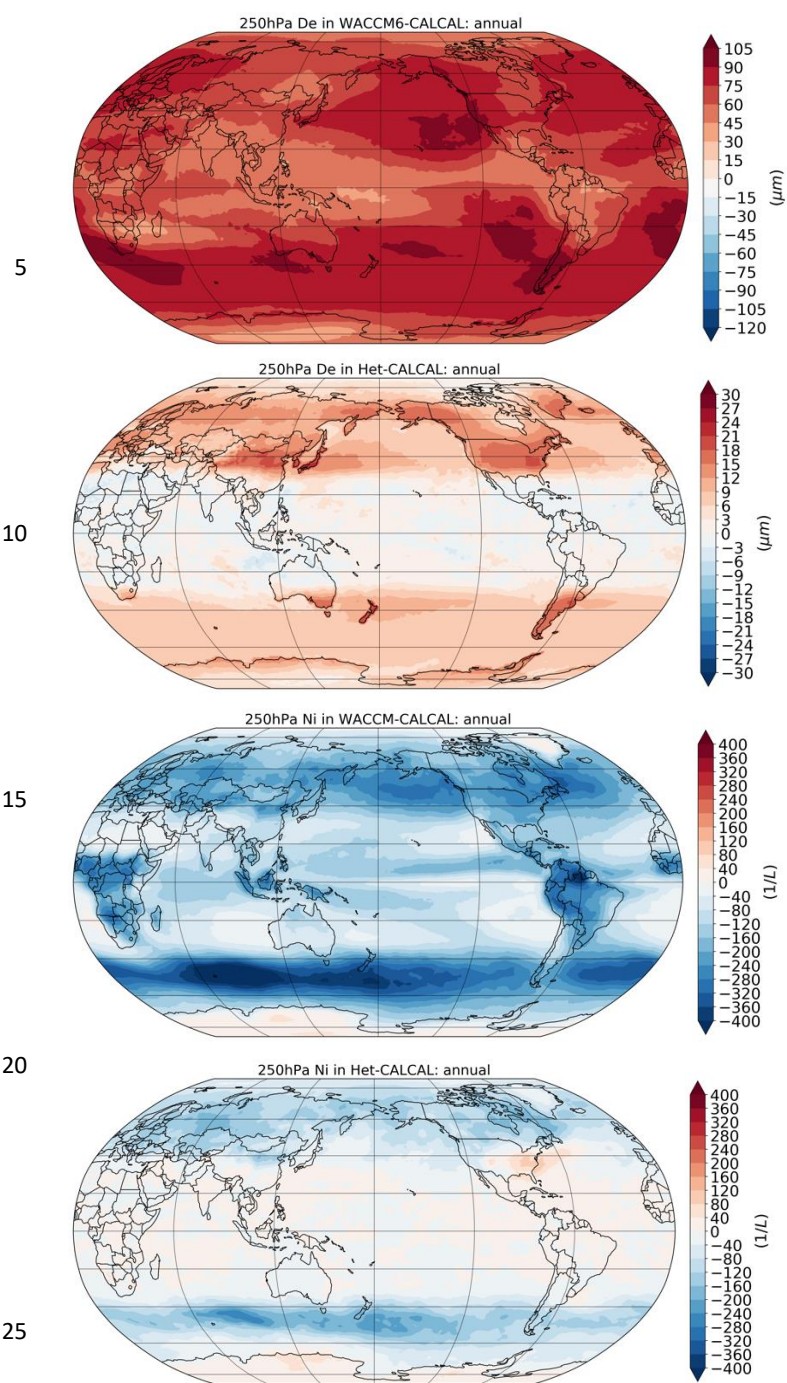

Figure 12. From top-to-bottom, WACCM6 – CALCAL, HET – CALCAL differences for D_e, and WACCM6 – CALCAL, HET – CALCAL differences for N, all at 250 hPa, based on the annual means in Figs. 10 and 11.





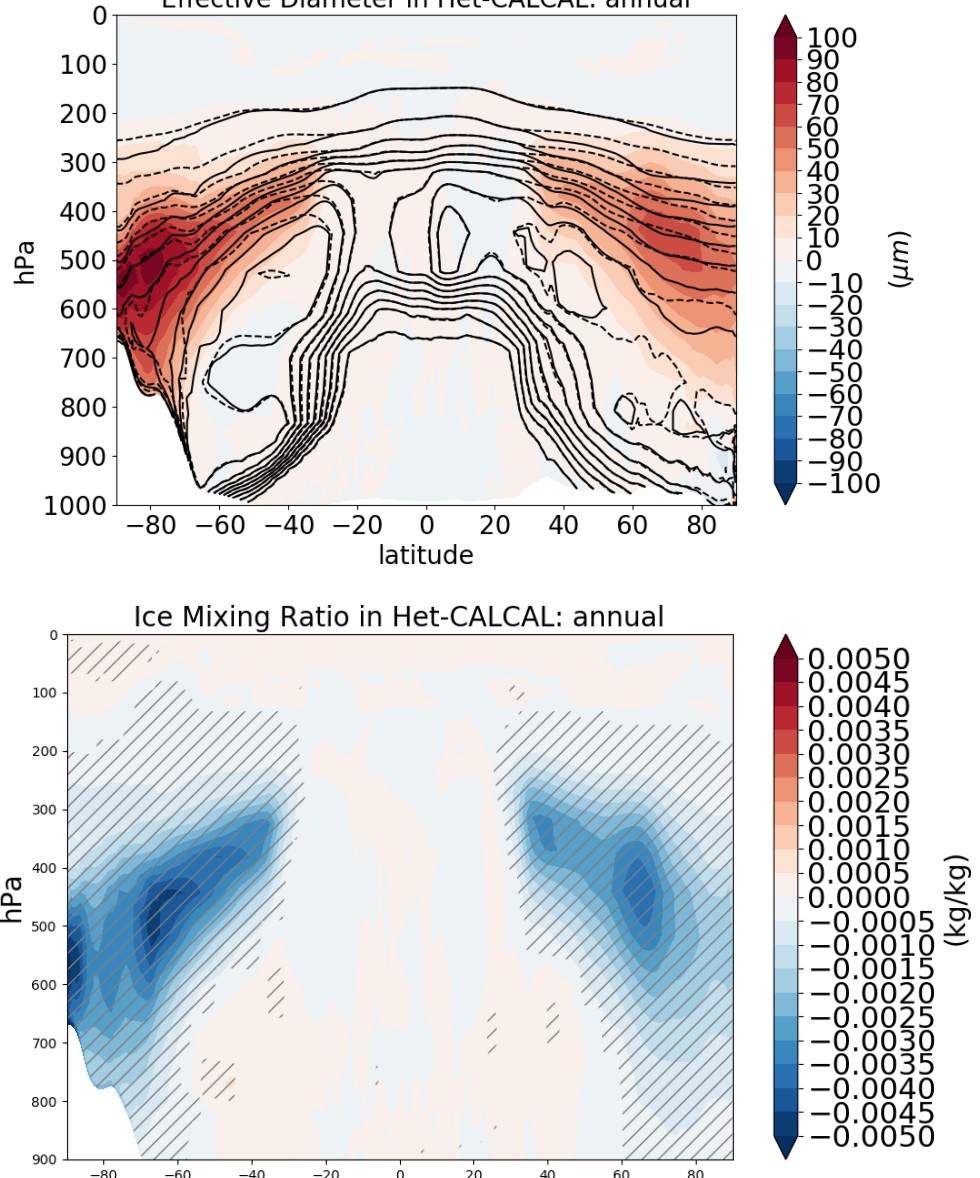

Figure 13. Upper: HET – CALCAL differences in zonal annual mean effective diameter ($D_e$). Solid contours are
for CALCAL $D_e$ while dashed contours are for HET $D_e$, each having 30 μm increments and ranging from 0 to 300
μm. The color contours represent HET – CALCAL $D_e$ differences. Lower: Zonal mean ice water content changes
are shown for different periods. The hatched areas were significantly different between HET and CALCAL with
95% confidence.

### Cloud Fraction in Het-CALCAL: annual

### RH in Het-CALCAL: annual

Figure 14. Upper: HET zonal mean changes (in percent) in cloud fraction relative to CALCAL. Lower: HET zonal
mean relative humidity changes (in percent) relative to CALCAL. The hatched areas indicate 95% confidence that
the differences are significant.



Figure 15. Annual mean and seasonal mean (DJF and JJA) HET – CALCAL differences in the cloud radiative effect (SWCF + LWCF) or CRE. Hatched areas are significant at the 95% confidence level.

Figure 16. Top-of-model annual mean and seasonal mean (DJF and JJA) HET – CALCAL difference for all-sky net flux. Hatched areas are significant at the 95% confidence level.

Figure B1.   Similar to Fig. 8 but with reversed axes.  In situ measurements of N, $A_{PSD}$ and IWC were calculated
from 2D-S probe PSD measurements (or estimates regarding IWC) taken during the SPARTICUS and TC4 field
campaigns.  These were used in Eqns. (12) and (28) to calculate $D_e$ and $R_v$ for each PSD, shown by the blue squares.
The red diamonds were determined theoretically for exponential PSD (see text for details).  The corresponding curve
fits estimate $D_e$ from in situ $R_v$ (or an estimate of retrieved $R_v$ from in situ $R_v$), and then estimate N from this $D_e$ and
the in situ climatology IWC.

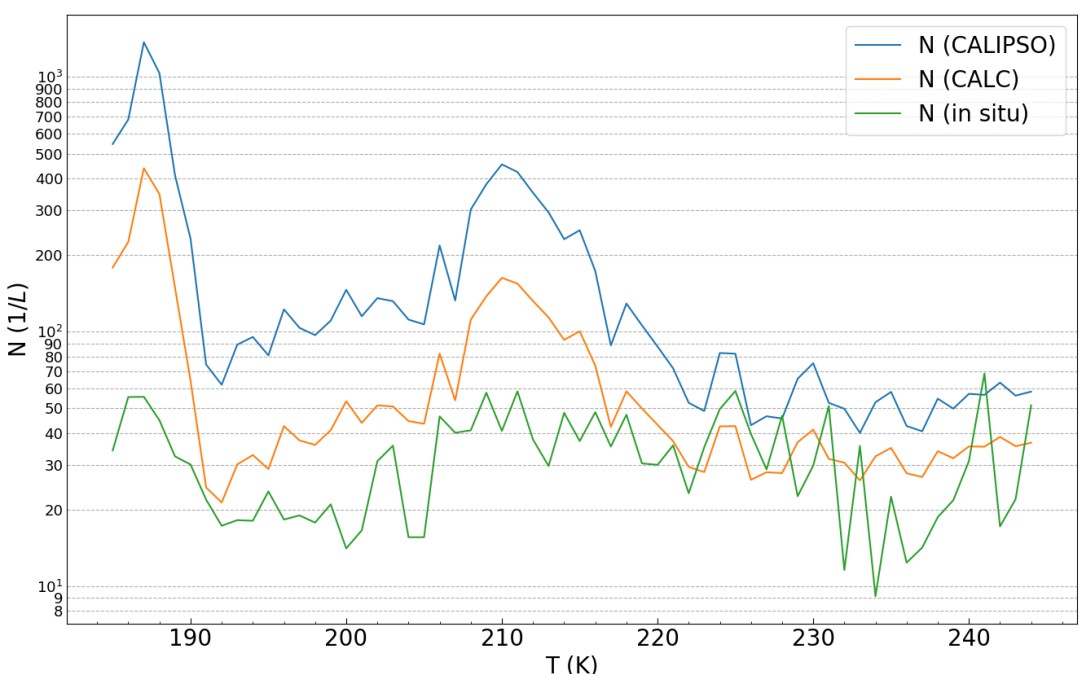

5    Figure B2.  Results of the study comparing N calculated from in situ $R_v$ and IWC (orange) with in situ N (green).
     Also shown is N calculated from $R_v$ representing CALIPSO retrievals averaged over the tropics and midlatitudes
     along with the in situ IWC (blue).