# Peer review of "An Estimate of Global, Regional and Seasonal Cirrus Cloud Radiative Effects Contributed by Homogeneous Ice Nucleation"

_Atmospheric Chemistry and Physics, 2020_

## Referee Comment (RC1) · Blaž Gasparini (Referee) · 26 Nov 2020

**Review of "Mitchell et al., 2020"**

This is an innovative work that uses CALIPSO satellite retrievals to derive an ice crystal effective diameter and ice crystal number dataset. This dataset is subsequently used as an input to the model microphysical scheme and together with the model-computed ice water content determines the ice crystal number concentration and the distribution shape parameter, bypassing the uncertainties associated with ice nucleation at cirrus conditions. The standard version of WACCM6 model is compared with the version driven by the CALIPSO effective diameter retrievals, to show substantial model deficiencies. Retrievals from the tropics are assumed to correspond to heterogeneously nucleated cirrus. A new model simulation that is supposed to represent a climate in which cirrus clouds are formed exclusively by heterogeneously nucleation, uses tropical effective diameter values at all latitudinal bands. The difference between the latter and the former simulation driven by standard CALIPSO-derived effective diameter, is argued to be a good proxy for radiative differences between a fully-heterogenous-cirrus-covered and reference climate.

I think it is valuable to find different approaches to address the magnitude of the change in cloud radiative effects (CRE) between a reference and a fully-heterogenous-cirrus-covered world, but at present I am not convinced that the way the newly developed innovative method is appropriate. I therefore hope that my numerous, but well-intentioned comments will help improve the study.
In particular, I am sceptical about the validity of the study's assumption that considers tropical cirrus data as a proxy for in-situ heterogeneous ice nucleation worldwide. The study has to therefore undergo a substantial review, before it can be accepted.

**General points:**

1) I think it's fairly certain that most of the tropical cirrus in the considered cloud optical depth (COD) range of 0.3 to 3 are composed of ice crystals from deep convective detrainment. In-situ cirrus in that latitudinal band typically form in the tropical tropopause layer and are optically very thin, falling below the lower threshold of the retrieval. The processes that contribute most to the ice crystals of tropical clouds in the COD range of 0.3 to 3 are very likely a combination of heterogeneous ice nucleation in mixed-phase conditions (a source of large ice crystals) and homogeneous nucleation of cloud droplets within convective updrafts (a source of large ice crystal number) (e.g. Krämer et al., 2020). I don't think in-situ ice nucleation can play a crucial role in determining anvil cloud microphysical and radiative properties. In presence of ice crystals, depositional growth is favoured over new ice nucleation events.
Therefore, I find it inappropriate to use tropical cloud properties as a proxy for in-situ heterogeneous ice nucleation.

I would find the results of the study more plausible if those were based on (or at least confirmed by) some extratropical cirrus data. It is plausible that extratropical locations over mountains on average represent cloud properties typical for homogeneous nucleation (HOM). Locations over oceans (or at least parts of oceans) and near the major dust sources may be more likely to represent cirrus clouds dominated by heterogeneous ice nucleation. The problem is that locations near the

major extratropical upper tropospheric dust sources are at the same time mountainous regions (areas surrounding the Taklamakan, Gobi deserts, Tibetan plateau etc.).

This is, however, not the only problem one would need to think about when using extratropical data as proxies for heterogeneous freezing, given the abundance of liquid-origin ice crystals, particularly in the storm track regions. Such ice crystals, similarly to those from deep convection, nucleate at warmer temperatures and are therefore a confounding factor for the presented analysis, that is focused on in-situ nucleation below the homogeneous freezing temperature of water. This problem needs to be therefore carefully addressed in the study.
Studies by e.g. Krämer et al., 2016, Wernli et al., 2016, Gasparini et al., 2018, Dietlicher et al., 2019 all show considerable contribution of liquid-origin cirrus at the warm part of the considered temperature range (until approximately -50°C or similar). One imperfect, but simple way to decrease the influence of liquid-origin ice crystals would be therefore to consider only the coldest of the cirrus clouds in this study. This would decrease the number of datapoints considered: but if the authors at the same time extend the analysis to the full CALIPSO dataset, the result may still be significant enough to produce a robust new version of a $D_e$ lookup table.

Finally, even in a situation where one would fix all the potential problems, the results would depend on the assumption that HET ice crystals have the same size in all regions of the world, only with a temperature dependence.  Which may be a reasonable first-order assumption due to the considered temperature dependence of the $D_e$ dataset/lookup table, but it still is only an assumption, and as such worth pointing out in the manuscript.

2) Ice crystal radius is a strong function of temperature due to the exponential dependence of deposition/sublimation (basically Clausius-Clapeyron relationship). Could the authors show whether the extratropical winter cirrus are occurring in a similar temperature range as summer cirrus? I.e. are the median temperature and its $25^{th}$ and $75^{th}$ quartiles comparable between different seasons? Can the seasonal cycle of temperatures be partially responsible for changes in ice radius and number?

3) Why are only 2 years of CALIPSO retrievals used in the $D_e$ calculations when the full dataset is 10+ years long? The figures showing CALIPSO retrievals are very noisy. Part of that may be physical, but I imagine that a lot of that noise will disappear by taking into consideration a longer dataset. I suggest the authors to consider doing so.

4) The following questions could be better addressed in the manuscript:
What fraction of cirrus is the retrieval used neglecting?
      Maybe the authors could estimate what fraction of CALIOP lidar profiles are
      fully attenuated in the selected temperature range?
What fraction of the total cirrus CRE is neglected?
      I believe the considered cirrus COD range may be OK, at least for the purpose
      of constraining in-situ cirrus. See more suggestions for useful literature
      particularly on the cirrus CRE in the specific comments section.

5) Could the authors discuss a potential drawback of the implemented method that arises due to applying the same ice crystal radius to all cirrus clouds of the same temperature and geographic location, not considering the likely possibility of a diversity between clouds of different COD and lifetime stage. One may expect the thinnest cirrus to have smaller radii, and the thickest, liquid-origin cirrus to have larger radii.

6) The hypothesis stating that deep convection and atmospheric mixing is responsible for the transport of most INPs in the upper troposphere is mentioned several times throughout the manuscript. While I think this is a plausible hypothesis, no evidence is provided that would support its validity (either from the authors' own work or by using appropriate citations).

I think there is evidence that convection, both moist and dry, can loft a substantial fraction of dust in summer in the broader Sahara region and in its outflow (e.g. Knippertz and Todd, 2011; Marsham et al., 2013; Van der Doest et al., 2018; etc.). Recent studies demonstrate the ability of deep convection in tropical Atlantic transporting dust in the upper troposphere, but do not quantify its climatologic effects (e.g. Twohy et al., 2017, Sauter et al., 2019), and refer only to the outflow of the Saharan desert.

Based on the current literature it is hard to believe convection is the dominant transport pathway of upper tropospheric dust in the extratropics or even further polewards. Note also: the largest part of the upper tropospheric dust can be traced back to the central Asian deserts, not Sahara (e.g. Fig. 3 in Hu et al., 2020; Xu et al., 2018; but also Uno et al., 2009). Several publications explain that by atmospheric advection/winds, and not convection. Moreover, it is interesting that those dust sources seem to peak in spring, and not in summer. Based on that fact one may expect rather a spring summer peak in heterogeneously nucleated cirrus (HET) in the northern hemisphere.

7) The article dedicates a lot of effort and space to verify the method of effective diameter retrieval. I believe the corresponding sections 2 and 4 should be shortened to give more space to the main topic of the paper, given that this is not a manuscript about a new method, but about the application of this method. Please find some suggestions in the specific comments.

**Specific comments**

Comment on title: The manuscript estimates the effect of HOM relative to HET (as stated in the abstract), and not the full radiative effect of HOM (which is what one would think based on the title).

Please be consistent: micron or µm (both are used interchangeably)

**Abstract:**

When writing northern or southern hemisphere the authors actually mean the band 30-90°N/S, I believe. This needs to be mentioned, as it makes a large difference!

What are the standard deviations of the mentioned radiative anomalies? Please add!

**Section 1: Introduction**

General comments:

1.) I miss a (short) paragraph reminding the reader about the climatic importance of ice crystal fall speed (in relation with HOM vs. HET cirrus).

2.) I also miss a paragraph describing the previous modelling results showing the relative importance between HOM and HET (e.g. Barahona et al., 2017, publications by Joyce Penner, publications using CAM model by Xiaohong Liu et al. (maybe Liu et al., 2012?), Gasparini et al., 2016, Muench and Lohmann, 2020, …)

3.) I would suggest using "$N_i$" instead of "$N$" for better clarity.

Specific comments:
Page 2, line 13: Please rephrase. Currently a reader would think that HET nucleation occurs at 100 % RHi, which is indeed not true, not even for the best INPs.

Page 2, lines 15-18: A sentence or two describing radiative implications of changes in $N_i$ and $D_e$ could be added to the introduction.

Page 2, line 19: Only some of the current models assume pre-existing ice. Not all. Not in all versions. E.g. to my knowledge cirrus seeding papers by Trude Storelvmo do not use pre-existing ice. Articles on cirrus formation by Barahona don't use pre-existing ice to my knowledge, at least the early papers. The new version of cloud microphysics in ECHAM-HAM does not use pre-existing ice any more (Muench and Lohmann, 2020). CAM5-CARMA model with sectional ice microphysics (e.g. Maloney et al., 2019) does not use pre-existing ice but rather the Liu et al., 2007 parameterization.
Only some models/model versions use the pre-existing ice notion. But that feature can be easily switched off if proved detrimental to the performance of the climate model.

It would be very valuable to add a new simulation performed with the pre-existing ice switch turned to "off". Please, consider adding it in the revised version of the

manuscript. That may directly confirm the hypothesis of the futility of pre-existing ice notion in this specific GCM.

Also, the idea of pre-existing ice dates back to Kärcher et al, 2006. Hendricks et al., 2011 was the first to implement it in a GCM, while also Kübbeler et al., 2014 implemented it in a GCM before the cited Shi et al., 2015.

Page 2, line 24: Results of the presented model simulations cannot be generalized to all models. HOM was quite widespread in several papers by Storelvmo et al. Pre-existing ice has been used in publications on cirrus clouds using ECHAM-HAM model (Kuebbeler et al., 2014, Gasparini et al., 2016,2017,2018,2020). However, the most recent ECHAM-HAM model version omits its use (see Muench and Lohmann, 2020).

Page 2, line 25: Zhao et al demonstrated…strong influence… What was the mechanism? Please add a few more words describing Zhao et al.'s results.

Page 2, line 30: see my general comment #6.

Page 2, line 31: How do anvil cirrus enhance HET? They suppress new ice nucleation events of all kinds, both HOM and HET. Indeed, they may suppress more HOM, if HOM would occur at all with no detrainment. Or do mean by transporting INPs to the upper troposphere?
"advected pre-existing ice": detrained (pre-existing) ice? That is, detrained from deep convective clouds, and not, e.g. ice crystals formed in a warm conveyor belt of a cyclone that are slowly advected to colder temperatures or similar.
Those ice crystals could simply be referred to as "detrained ice crystals", as that's what those are called both in modeling world and beyond it?
As mentioned in my general comment #1, I believe anvils contain mainly detrained ice crystals. It is hard to nucleate new ice crystals, in presence of numerous detrained ice crystals (even HET). Current best knowledge of anvil microphysics does not assume an important role of new nucleation. It may be present, but not crucial for anvil evolution (e.g. Jensen et al. 2009, Gasparini et al., 2019, Wall et al., 2020).

Some support for potential new ice nucleation may come from results of Sokol and Hartmann 2020. Also from a modelling study by Hartmann et al., 2018: however this one was done in a highly idealized, fully cloud-covered domain and their indication of the importance of newly nucleated ice crystals for the maintenance of anvils have to be taken with some caution.

**Section 2: Methodology**

General comments:

1.) Please use appropriate formatting for the equations (in particular eq. 1, 16,27)

2.) What is the size of snow in the newly modified scheme? How are ice and snow split in the modified microphysics?

3.) Is the lookup table for $D_e$ applied same both to snow and ice categories?

Section 2.2
Page 7, lines 20-30: the fact that Mitchell (1996) fall speeds are within 10% of Heymsfield and Westbrook (2010) is mentioned twice.

Is the manuscript really benefiting from all the detailed information from the Section 2.2 (particularly the last part of this section)? I would suggest shortening it to only what is strictly needed for understanding the result section (while the rest could go in the appendix).

What should a reader take out of Fig. 4? I am confused as it states on page 7 that the terminal velocity from Mitchell et al. (1996) is within 10% of the Heymsfield and Westbrook (2010). The plot, on the other hand, shows a significantly larger deviation. Would it be fair to comment that there is substantial uncertainty in all of such estimates? Could the authors give us a hint of a plausible range of sedimentation velocities instead (rather than just numbers)?

General comments about the method that refer mainly to Sections 2.3 and 2.4

I) Is the new microphysics driving the $D_e$ and consequently $N_i$ of ice only, or also of snow? How is snow number and size affected by the new microphysics?

II) Despite a very detailed description of the method of selection of $D_e$ and how this is integrated in the model, I am missing a practical example from the model simulation. Practical examples will help readers understand the described method in a quicker and more intuitive way than a combination of text and equations.

For example, a typical example of a tropical and an extratropical cirrus could be mentioned explicitly – maybe one with high and one with low IWC values, e.g.

  1.) high IWC; T= X;  p = Y => what is the ICNC and $D_e$?
  2.) low IWC;  T = X;  p = Y => what is the ICNC and $D_e$?

III) The fact that sublimation and deposition do not change the $D_e$ in the simulation is counterintuitive and should therefore be better highlighted in the manuscript. What is the range of variability in $N_i$ that results out of changes in IWC due to sublimation/deposition?

Section 2.3:

Page 9, lines 1-5: I am still not sure whether "snow" is treated exactly the same as "ice" (that is, using the same $D_e$).

Page 9, lines 24-30: Is this paragraph really needed? In my opinion it could be moved to the appendix. Does Fig. 5 need to be shown in the main part of the manuscript? Keep in mind it is not a manuscript about a method, but about a radiative effect of changes between HOM and HET freezing.

Page 9, line 26: What is "mks"?

Page 9, lines 30-32 and line 1 on page 10: Is this an important information? What should a reader take from it?

Section 2.4:

Page 10, equation 27: Please change the formatting; it is really hard to understand it in its present shape

Page 10, lines 11-23: Do the readers really need the information provided on lines 11-23, particularly given that accretion is not affected by the changes in microphysics?

**Section 3: Experimental design**

Page 11, lines 10-15:  As I asked in general comment #4, this point needs to be better justified.
Suggested literature: Hong and Liu, 2016, Matus and L'Ecuyer, 2017, Berry and Mace, 2014, Berry et al., 2019

Page 11, lines 20-25: This has also been addressed already in a question above. I think it's hard to argue that most of tropical clouds retrieved by the described method would be composed prevalently of particles that freeze at temperatures colder than the homogeneous freezing temperature of water.

Page 12, lines 10-23: This paragraph does not talk about experimental design and has to therefore be removed or find place elsewhere.

Page 12, lines 28-31: Please rephrase the sentence starting with "Ice-supersaturated regions…" as it is very convoluted in its current form.

**Section 4: Comparisons of CALIPSO retrievals with an in situ cirrus cloud climatology**

This sections could be significantly shortened or removed from the main part of the manuscript. It does provide valuable information, but is not central to the main messages of the manuscript.

Data published in Krämer et al., 2020 do not show an increased ice radius in the tropics compared to midlatitudes. Why do they apparently disagree with the used CALIPSO dataset?
Why is the larger ice crystal radius in tropics in comparison to extratropics not observed in the Krämer et al., 2020 data?

Why do results from Krämer et al., 2020 suggest the largest $N_i$ in the tropics, followed by midlatitudes, and finally high latitudes? This is in contrast to the CALIPSO dataset.

Figure 9: Why do the extratropical $R_{ice}$ differ significantly from the tropical one only in the warmer part of the temperature range? It would also help understanding the significance of such changes by plotting the interquartile range for CALIPSO data.

**Section 5: Modeling results and discussion**

Figures 10 and 11: I think it would be fairer to show $D_e$ and $N_i$ at a constant temperature level, and not pressure level. Indeed, the $D_e$ will be larger in the tropics simply due to ~10 K warmer temperatures compared with the extratropics. The authors could, for instance, show the properties in a temperature bin between -50 and -60°C.

Page 15, lines 24-27: I think the high $N_i$ in the storm tracks may be related to a larger contribution of liquid-origin cirrus relative to in-situ cirrus.

Section 5.1

I would suggest to first describe the main and expected results related to cirrus, and only after than the unexpected and likely very model dependent changes of mixed-phase clouds.

Figure 13: Currently, the reader is overwhelmed by a large number of contour lines. Please consider plotting 3 panels for each quantity showing separately results from: Het, CALCAL and Het-CALCAL. It would be also useful if 235 K and 273 K isotherms are added on all subpanels. Those lines will help guide the eye to better distinguish regions of cirrus from those of mixed-phase clouds.

Section 5.2

Table 3: Could SW and LW CRE anomalies be shown separately and briefly commented in the text?

It would be great if the manuscript would provide also a decomposition of the CRE and CRE anomalies based on temperature range of separate cloud species (e.g. with the help of a double call to the radiation routine). This is easy to do in a CAM-like model, without the need to change model code.
What part of the net CRE changes are related to changes in cirrus cloud only?
What part to changes in mixed-phase clouds?

Note that Gruber et al., 2019 also observed significant changes to mixed-phase clouds following cirrus cloud seeding. Some effects to the mixed phase were in addition presented by Gasparini et al., 2017 (please refer to their Table 5 and the relevant discussion). Could their results be compared with the changes observed in mixed-phase regime in this study?

Page 17, lines 20-22: The sentence starting with: "At lower latitudes…" is incorrect. The warming effect by clouds at T<-35°C is strongest in tropics. It is true that the SW effects in the tropics are stronger due to larger insolation, but LW effects are stronger too due to colder temperatures of the peak in upper tropospheric cloud fraction. Please refer to:
1. observational studies:

Hong and Liu, 2016 (particularly their Fig. 6) and Matus and L'Ecuyer, 2017 (particularly their Fig. 5), Kubar et al, 2007, Gasparini et al., 2019 (particularly their Fig. 1)
2. modelling studies:
Gasparini et al., 2017 (particularly their Fig. 3), Gasparini et al., 2020 (particularly their Fig. 1), Muench and Lohmann, 2020 (their Fig. 13)

It is true, however, that if we consider ALL clouds, including those at warmer temperatures, the CRE in regions dominated by tropical deep convection is near neutral (e.g. Wall et al., 2019).

I am also attaching my own figure produced with a cloud resolving model (horizontal resolution of 1 km, vertical resolution of 250 m in the upper troposphere, RRTMG radiative transfer model) for mean tropical insolation to show how clouds in the COD range of 0.3 to 3 over tropical oceans (resembling conditions in the Western Pacific) on average lead to positive net CRE anomalies at the top-of-the-atmosphere (in red: LW CRE, in blue: -SW CRE, in black: net CRE).

[Figure]

Section 5.3

Page 18, line 2: RH radiative effect: Why RH? Isn't it the water vapor mixing ratio that matters for the radiative transfer calculations, and not the relative humidity?
So it's the water vapor top-of-the-atmosphere radiative effect.

Page 18, lines 1-13: The authors are talking and speculating about the clear sky radiative effect changes. Why speculating – just show the clear sky radiative anomalies in a table (ΔRadiation_clear = ΔRadiation_full - ΔCRE)!

Page 18, lines 15-16: "…it is less likely that ozone and other greenhouse gas…significantly changing…".
Is it changing or it is not changing? The model output will give the answer.

Page 18, line 19: Same for the aerosols – are they changing significantly or not?
Aerosol optical depth may give some hints about it. More importantly than scavenging, dynamical changes may change sources and advection of aerosols.

Page 18, line 23: "…IWC could be higher…" Is it higher or not? Please find the answer in the model output fields, e.g. by comparing ice water path values.

**Conclusions**

The first few paragraphs of conclusions are pretty vague, just reiterating some generally accepted knowledge that what mentioned earlier in text that fits better in the introduction.
Also, as mentioned earlier, one should be very cautious when mentioning the hypothesis of deep convection increasing INP concentration.

Page 18, line 31: "…pre-existing ice advected into anvil cirrus…"
In my opinion, detrained ice crystals equal anvil clouds (as mentioned earlier, I wouldn't necessarily call them pre-existing ice). They are not detrained/advected into anvils, they are anvils.

Page 19, lines 4-9: This paragraph is extremely speculative and would better fit in the final part of the conclusion.
Why should there be a substantial increase in upper tropospheric INPs with global warming? Is there some evidence for it?
Keep in mind the total aerosol indirect effect of dust on clouds may be very close to zero, at least in global average. An increase in INP decreases cirrus coverage to release more radiation back to space. On the other hand, the effect on mixed-phase clouds would lead to the opposite, warming effect, glaciating more clouds, decreasing their reflective properties. This may be different in high latitude winter. A recent study by McGraw et al., 2020 provides a nice overview.

Page 19, line 10: The first sentence isn't totally correct: those studies were attempting to estimate the radiative effect of a change from HOM to HET, not the full HOM radiative effect. I also don't believe simulations using increased sedimentation velocity as a proxy for seeding can be put in the same box. It would be good to split the long list of references to distinguish those two groups of studies.

Page 19: Also worth mentioning that what this study is estimating is the maximum effect of a change between HOM and HET. So we may need to consider that as the upper bound of seeding in a globally uniform sense, at least.

Page 19, lines 25-28:The results of this study cannot be generalized to all models. This may be true for the model version used in this study, with a specific set of tuning parameters that produced those clouds. Apparently, the WACCM6 model development team did not care too much about the representation of cirrus clouds, which were left significantly out of the range of used observations (if we assume the CALIPSO retrieval as truth).
Unfortunately, tuning parameters that are not constrained by observations can substantially determine a lot of cloud and climate properties in climate models. So it may not always be only a question of HOM vs. HET in the modeling world.

Finally, Fig. 2k in Gasparini et al., 2018 shows that the IC radius at cirrus levels seems probably closer to the used CALIPSO observations compared to WACCM6. An interesting comparison between CESM1 (CAM5) with Barahona and Nenes, 2008

freezing and ECHAM-HAM is moreover shown in the supplementary figure 1 of Gasparini et al., 2020.  CESM1 shows numerous small ice crystals in the upper troposphere, unlike the version of the WACCM model used in the present study. So while the statement is correct for WACCM6, it cannot be generalized to all climate models.

Page 19, line 28: The dynamical impact of atmospheric cloud radiative effects (ACRE) by high clouds (i.e. radiative heating within the atmospheric column) has been substantially studied in the past 10 years by several authors (including, very prominently, the cited Ying Li). I suggest some more potentially relevant citations: Voigt et al., 2019, Voigt and Shaw, 2016, Albern et al., 2019, Watt-Meyer et al., 2017, etc.

**Reference:**

- Albern et al., 2019: Cloud-Radiative Impact on the Regional Responses of the Midlatitude Jet Streams and Storm Tracks to Global Warming, JAMES, doi: 10.1029/2018MS001592
- Barahona and Nenes, 2009: Parameterizing the competition between homogeneous and heterogeneous freezing in cirrus cloud formation - monodisperse ice nuclei, ACP, doi: 10.5194/acpd-8-15665-2008
- Barahona et al., 2017: Direct estimation of the global distribution of vertical velocity within cirrus clouds, Sci. Rep., doi: 10.1038/s41598-017-07038-6
- Berry and Mace, 2014: Cloud properties and radiative effects of the Asian summer monsoon derived from A-Train data, JGR-A, doi: 10.1002/2014JD021458
- Berry et al., 2019: Using A-Train observations to evaluate cloud occurrence and radiative effects in the Community Atmosphere Model during the Southeast Asia summer monsoon, JClim, doi: 10.1175/JCLI-D-18-0693.1
- Dietlicher et al., 2019: Elucidating ice formation pathways in the aerosol-climate model ECHAM6-HAM2, doi: 10.5194/acp-19-9061-2019.
- Gasparini et al., 2016: Why cirrus cloud seeding cannot substantially cool the planet, JGR-A, doi: 10.1002/2015JD024666
- Gasparini et al., 2017: Is increasing ice crystal sedimentation velocity in geoengineering simulations a good proxy for cirrus cloud seeding?, ACP, doi: 10.5194/acp-17-4871-2017
- Gasparini et al., 2018: Cirrus cloud properties as seen by the CALIPSO satellite and ECHAM-HAM global climate model, JClim, doi: 10.1175/JCLI-D-16-0608.1
- Gasparini et al., 2019: What drives the lifecycle of tropical anvil clouds?, JAMES, doi: 10.1029/2019MS001736
- Gasparini et al., 2020: To what extent can cirrus cloud seeding counteract global warming?, ERL, doi: 10.1088/1748-9326/ab71a3
- Gruber et al., 2019: A Process Study on Thinning of Arctic Winter Cirrus Clouds With High-Resolution ICON-ART Simulations, JGR-A, doi: 10.1029/2018JD029815
- Hartmann et al., 2018: The Life Cycle and Net Radiative Effect of Tropical Anvil Clouds, JAMES, doi: 10.1029/2018MS001484

- Hendricks et al, 2011: Effects of ice nuclei on cirrus clouds in a global climate model, JGR-A, doi: 10.1029/2010JD015302
- Hong and Liu, 2016: Assessing the Radiative Effects of Global Ice Clouds Based on CloudSat and CALIPSO Measurements, JClim, doi: 10.1175/JCLI-D-15-0799.1
- Hu et al., 2020: Modeling dust sources, transport, and radiative effects at different altitudes over the Tibetan Plateau. ACP, doi: 10.5194/acp-20-1507-2020
- Jensen et al., 2009: On the Importance of Small Ice Crystals in Tropical Anvil Cirrus, ACP, doi: 10.5194/acp-9-5519-2009
- Kärcher et al., 2006: Physically based parameterization of cirrus cloud formation for use in global atmospheric models, JGR, doi: 10.1029/2005JD006219
- Knipperz and Todd, 2011: Mineral Dust Aerosols over the Sahara: Meteorological Controls on Emission and Transport and Implications for Modeling, Reviews of Geophysics, doi:10.1029/2011RG000362
- Kuebbeler et al., 2014: Dust ice nuclei effect on cirrus clouds, ACP, doi: 10.5194/acp-14-3027-2014
- Krämer et al., 2016: A microphysical guide to cirrus clouds – Part 1: cirrus types, *ACP,* doi: 10.5194/acpd-15-31537-2015
- Krämer et al., 2020: A Microphysics Guide to Cirrus – Part II : Climatologies of Clouds and Humidity from Observations, *ACP,* doi: 10.5194/acp-2020-40
- Kubar et al., 2006: Radiative and convective driving of tropical high clouds, JClim, doi: 10.1175/2007JCLI1628.1
- Liu et al., 2007: Inclusion of ice microphysics in the NCAR Community Atmospheric Model version 3 (CAM3), JClim, doi: 10.1175/JCLI4264.1
- Liu et al., 2012: Sensitivity studies of dust ice nuclei effect on cirrus clouds with the Community Atmosphere Model CAM5, ACP, doi: 10.5194/acp-12-12061-2012
- Maloney et al., 2019: An Evaluation of the Representation of Tropical Tropopause Cirrus in the CESM/CARMA Model Using Satellite and Aircraft Observations, JGR-A, doi: 10.1029/2018JD029720
- Marsham et al., 2013: Meteorology and dust in the central Sahara: Observations from Fennec supersite-1 during the June 2011 Intensive Observation Period, JGR, doi:10.1002/jgrd.50211
- Matus and L'Ecuyer, 2017: The role of cloud phase in Earth's radiation budget, *JGR-A,* doi: 10.1002/2016JD025951
- McGraw et al., 2020: Global radiative impacts of mineral dust perturbations through stratiform clouds, JGR-A, doi: 10.1029/2019JD031807
- Muench and Lohmann, 2020: Developing a Cloud Scheme With Prognostic Cloud Fraction and Two Moment Microphysics for ECHAM-HAM, JAMES, doi: 10.1029/2019MS001824
- Sauter et al., 2019: The Observed Influence of Tropical Convection on the Saharan Dust Layer, JGR-A, doi: 10.1029/2019JD031365
- Sokol and Hartmann, 2020: Tropical Anvil Clouds: Radiative Driving Towards a Preferred State, JGR-A, doi: 10.1029/2020JD033107
- Twohy et al., 2017: Saharan Dust, Convective Lofting, Aerosol Enhancement Zones and Potential Impacts on Ice Nucleation in the Tropical Upper Troposphere, JGR-A, doi: 10.1002/2017JD026933

- Van der Does et al, 2018: The mysterious long-range transport of giant mineral dust particles, Science Advances, doi: 10.1126/sciadv.aau2768
- Voigt and Shaw, 2016: Impact of regional atmospheric cloud radiative changes on shifts of the extratropical jet stream in response to global warming, JClim, doi: 10.1175/JCLI-D-16-0140.1
- Voigt et al., 2019: The atmospheric pathway of the cloud-radiative impact on the circulation response to global warming: Important and uncertain, JClim, doi: 10.1175/JCLI-D-18-0810.1
- Watt-Meyer et al., 2017: Local and Remote Impacts of Atmospheric Cloud Radiative Effects Onto the Eddy-Driven Jet, GRL, doi: 10.1002/2017GL074901
- Wall et al., 2019: Is the Net Cloud Radiative Effect Constrained to be Uniform Over the Tropical Warm Pools?, GRL, doi: 10.1029/2019GL083642
- Wall et al., 2020: Observational Evidence that Radiative Heating Modifies the Life Cycle of Tropical Anvil Clouds, JClim, doi: 10.1175/JCLI-D-20-0204.1
- Wernli et al., 2016: A trajectory-based classification of ERA-Interim ice clouds in the region of the North Atlantic storm track, GRL, doi: 10.1002/2016GL068922.
- Xu et al., 2018: Tibetan Plateau Impacts on Global Dust Transport in the Upper Troposphere, JClim, doi:10.1175/JCLI-D-17- 0313.s1

---

## Short Comment (SC1) · 16 Dec 2020

The body text contains very useful discussion of CCT. This should be summarised in the abstract.

---

## Referee Comment (RC2) · Anonymous Referee #3 · 19 Jan 2021

Review of "An Estimate of Global, Regional and Seasonal Cirrus Cloud Radiative Effects Contributed by Homogeneous Ice Nucleation", by David L. Mitchell, John Mejia, Anne Garnier, Yuta Tomii, Martina Krämer, and Farnaz Hosseinpour, acp-2020-846.

Is homogeneous ice nucleation important in in-situ generated cirrus (temperatures below $-38^0$C)? Here's my thought on the question. Yes, in orographic wave clouds, in gravity waves, and in small convective cells that lead to cirrus uncinus clouds. Is homogeneous ice nucleation important globally? How would one address this question?

This study, which is densely packed and comprehensive, attempts to address this question through use of CALIOP lidar data and the using 40-year simulations from the NCAR Whole Atmosphere Community Climate Model version 6 (WACCM6). The model microphysics used is version 2 of the Morrison-Gettelman scheme (MG2). The effective diameter used as a reference is from the lidar. To estimate the heterogeneous (het) component, a somewhat questionable criteria is used.  The CALIPSO De – T relationships obtained from the tropics are applied globally. For a given T, the De for latitude zones 0 – 30 N and 0 – 30 S are averaged for each season, and the resulting look up tables for ± 30° latitude are applied to all six 30° latitude zones. The assumed that because effective diameters were highest and concentrations lowest in this region compared to outside of it, the averaged values for this region would be most representative of het nucleation. My question is whether observations of in-situ cirrus in this zone support this view. I would like the authors to address this question. Possibly another way to do this would be to turn off homogeneous ice nucleation (hom) in the model and to assess the resulting changes. I look at the MG and MG2 microphysics articles and it was unclear to me exactly how hom was calculated. Although I don't know a better way for the authors to have approached this problem, I do question the reliability of the results. I do like the CALCAL method, which uses the CALIPSO De retrievals to scale the WACCM6 model.

Three other significant points I'd like to mention. First, in Sections 2.1-2.3, why are these formulations necessary-why couldn't they just be derived from the SPARTICUS/TC4 data sets? Secondly and more significant, the data sets you use for much of the analysis are extremely limited geographically and seasonally. Why not use the Kramer et al. (2020) for your analysis? Lastly, might it be possible to add a figure that graphically shows the results from Figure 10-16?

Other points of mention:

Page 1, lines 10 Specifically mention "in situ generated cirrus"

1, 10-11 How about anvils that shear off-they are likely the result of ice crystals nucleated at warmer temperatures?

2, line 2 In-situ generated

2, 13-14. How is De derived from CALIPSO? Brief mention of it would be useful, and perhaps an estimate of the error.

5, Eq (7). Why not do this directly from the data?

Eq. (17) I can see why you would like to use c and d based on Mitchell and Heymsfield but I suggest using the more recent values derived by Heymsfield and Westbrook (2010). I see you mention the differences later but wonder why your choice.

8, 25. But Vn compares poorly between this scheme and Heymsfield and Westbrook (2010). Perhaps this should be mentioned.

Eq. (25). Can't Dmean be expressed as a function of the IWC?

9, paragraph beginning on line 24. I'm concerned that the Dmean values of 50 microns have serious errors, because the 2DS probe data shouldn't really be used below about 30 microns, and so the estimates are likely in error because all of the 2DS sizes are used to derive Dmean. I do see that you also do the calculations for Dmean beginning in the second bin but still there are some questions that I have about the uncertainty (depth of field, etc.)

10, Section 2.5. I really like the method used for smoothing.

11, 12. I thought an optical depth range up to 2.0 was valid but not to 3.0.

12, 28. Sourdeval et al. (2018) in their Part I use a much larger data base than you use. They use COALESC, ML-CIRRUS , ACRIDICON-CHUVA, ATTREX-2014 and SPARTICUS. I wish your data set was not based only on SPARTICUS and TC4.

12, 30. Any high RHi region would be short-lived.  Can this really be concluded from satellite-based retrievals?

12, 31. After doing digging, I went to the Petzold et al. (2020) article. They used MOZAIC (1994–2010) data as well as other data sets. You should mention that these RHi were derived from measurements on commercial aircraft.

15, lines 20-27. How much is CALCAL in the tropics due to deep convection?

16, lines 2-7. The problem with CALCAL in the mixed-phase zone is that it's unreliable because the CALIOP lidar is probably partially or fully occulted in that zone.

18, 25-27. Are there indications from in-situ data that support the view that high ice concentrations (10's per cm3) are found in updraft regions of cirrus (not mountain waves) at temperatures below -38C at which hom might be expected?